# Modal shift in North Atlantic seasonality during the last deglaciation

Geert-Jan A. Brummer[1,2], Brett Metcalfe[2,3*], Wouter Feldmeijer[2,4], Maarten A. Prins[2], Jasmijn van 't Hoff[2,5], Gerald M. Ganssen[2]

[1]NIOZ Royal Netherlands Institute for Sea Research, Department of Ocean Systems, 1790 AB, Den Burg, and Utrecht University, The Netherlands
[2]Earth and Climate Cluster, Department of Earth Sciences, Faculty of Sciences, VU University Amsterdam, de Boelelaan 1085, 1081 HV, Amsterdam, The Netherlands
[3]Laboratoire des Sciences du Climat et de l'Environnement, LSCE/IPSL, CEA-CNRS-UVSQ, Université Paris-Saclay, F-91191 Gif-sur-Yvette, France
[4]Now at: Nebest B.V., Marconiweg 2, 4131 PD, Vianen, The Netherlands
[5] Now at: Institute of Geology and Mineralogy, University of Cologne, Zuelpicher Str. 49a, 50674 Cologne, Germany

*Correspondence to*: Brett Metcalfe (b.metcalfe@vu.nl)

**Abstract.** Change-over from a glacial to an interglacial climate is considered as transitional between two stable modes. Palaeoceanographic reconstructions using the polar foraminifera *Neogloboquadrina pachyderma* highlight the retreat of the polar front during the last deglaciation in terms of both its decreasing abundance and stable oxygen isotope values ($\delta^{18}O$) in sediment cores. While conventional isotope analysis of pooled *N. pachyderma* and *G. bulloides* shells show a warming trend concurrent with the retreating ice, new single shell measurements reveal that this trend is composed of two isotopically different populations that are morphologically indistinguishable. Using modern time-series as analogues for interpreting down-core data, glacial productivity in the mid North Atlantic appears limited to a single maximum in late summer, followed by the melting of drifting icebergs and winter sea ice. Despite collapsing ice sheets and global warming during the deglaciation a second 'warm' population of *N. pachyderma* appears in a bimodal seasonal succession separated by the subpolar *G. bulloides*. This represents a shift in the timing of the main plankton bloom from late to early summer in a 'deglacial' intermediate mode that persisted from the glacial maximum until the start of the Holocene. When seawater temperatures exceeded the threshold values, first the "cold" (glacial) then the "warm" (deglacial) population of *N. pachyderma* disappeared, whilst *G. bulloides* with a greater tolerance to higher temperatures persisted throughout the Holocene to the present day in the mid-latitude North Atlantic. Single specimen $\delta^{18}O$ of polar *N. pachyderma* reveal a steeper rate of ocean warming during the last deglaciation than appears from conventional pooled $\delta^{18}O$ average values.

## 1. Introduction

### 1.1 Seasonality and single foraminiferal analysis (SFA)

Stable oxygen isotopes ($\delta^{18}O$) of pooled foraminifera have been used as key tracers of water masses (e.g., Epstein and Mayeda, 1953; Frew et al., 2000), ice-volume and sea-level fluctuations (e.g., Grant et al., 2012; Waelbroeck et al., 2002; Shackleton, 1987) over glacial-interglacial cycles (e.g., Pearson, 2012; Waelbroeck et al., 2005). Technical advances allow (Killingley, et al., 1981; Schiffelbein and Hills, 1984; Oba, 1990, 1991; Billups and Spero, 1996) for the routine analysis of the stable isotopic composition of single microscopic shells (Feldmeijer et al., 2015; Lougheed et al., 2018; Metcalfe et al., 2015; Metcalfe et al., 2019; Frizt-Endres et al., 2019) and chambers (Takagi et al., 2015; 2016; Lougheed et al., 2018; Pracht et al., 2018) of foraminifera permitting the resolution of (sub-)seasonal contrasts in seawater temperature (Ganssen et al., 2011; Wit et al., 2010; Groenveld et al., 2019), in lieu of pooled specimens that capture an averaged state of the

system on longer time scales. The use of single shell oxygen isotope analysis allows for moving beyond the "average" state of the climate system as expressed in pooled specimen analysis to observe the inter-specimen variance (e.g., Leduc et al., 2009; Koutavas et al., 2006; Koutavas and Joanides, 2012; Scussolini et al., 2013) that includes seasonal differences (Feldmeijer et al., 2015; Ganssen et al., 2011; Metcalfe et al., 2015; Wit et al., 2010). Seasonal changes during these glacial-interglacial cycles have rarely been addressed although resolving seasonal contrasts would significantly improve our understanding of past climate change (Huybers, 2006; Schmittner et al., 2011).

## 1.2 Aims and Objectives

As the largest ocean carbon sink in the northern hemisphere, the North Atlantic Ocean (Gruber et al., 2002) exhibits strongly seasonal productivity in the Present-Day. Deep wind-driven mixing in winter resupplies the photic zone with nutrients brought up from subsurface depths (Falkowski and Oliver, 2007) leading to phytoplankton blooms and maxima in the abundance of zooplankton including planktonic foraminifera during the onset of summer stratification, followed by a decrease as oligotrophic summer conditions develop. Present day temperature conditions in the mid-latitude North Atlantic (Fig. 1) preclude the occurrence of *Neogloboquadrina pachyderma* (Kretschmer et al., 2016), the species being restricted to the (sub)polar water masses in the high-latitude North Atlantic (Kohfeld et al., 1994). With the southward shift of the polar front that accompanied the last glacial, more favourable conditions developed, whereas *Globigerina bulloides* (Ganssen and Kroon, 2000) existed throughout. Here we analyse single shell stable oxygen and carbon isotopes of the planktonic foraminifera left-coiling *N. pachyderma* (e.g., Mekis et al., 2019; Metcalfe et al., 2019) and *G. bulloides* (e.g., Ganssen et al., 2011; Metcalfe et al., 2015; Sadekov et al., 2016) in a sediment core from the Iceland Basin in the mid-latitude Atlantic in order to address the direction of mean annual temperature change and seasonal changes during the past deglaciation. Given that present conditions in the mid North Atlantic are an anathema to the polar species *N. pachyderma*, this species is only used during the glacial and deglacial sections of the core.

## 2. Methodology

Piston core T88-3P (56.49°N, 27.80°W; Figure 1) was taken on the eastern flank of the Mid-Atlantic Ridge during the 1988 RV *Tyro* expedition of the Actuomicropalaeontology Palaeoceanography North Atlantic Project (APNAP) II. Piston core T88-3P measures 937 cm in length (Figure's 2 to 4) and was retrieved from above both the modern and glacial CCD (core water depth: 2819 m) ensuring minimal bias by carbonate dissolution. Core sections were manually split into a working half and an archive half.

## 2.1 X-Ray fluorescence core scanning and composite images

Archive halves of each section of the entire piston core were analysed at 1-cm down-core resolution using the Avaatech XRF core scanner (Richter et al., 2006), at the Royal NIOZ (Figure 4). Optical line-scanning was first performed on the split halves allowing a detailed and accurate description of visual and chromatic changes in core texture (Supplementary Figure 1). Prior to XRF-analysis the surface of the archive halves was scraped cleaned and each section was covered in SPEXCerti Ultralene® ultra-thin (4 μm) film. Bulk chemical composition was measured using energy dispersive fluorescence radiation, as elemental intensities in counts per second (CPS) at 10 kV (for 10 seconds) and at 50 kV (for 40 seconds). Despite limitations upon the accuracy and precision (Weltje and Tjallingii, 2008) by matrix effects, sediment (e.g., water content; grain-size) and measurement properties (e.g. surface irregularities) as well as machines settings used

(outlined above), the reliability for the elements Ca and Ti used herein is well established (Weltje and Tjallingii, 2008). To further minimize error, counts are expressed as log-ratios of two elements (Weltje and Tjallingii, 2008). Herein the Log (Ca/Ti) is used as a proxy for two end-members: marine productivity ([Ca]) and detrital terrestrial material ([Ti]) with minor contribution to [Ca] via detrital carbonate, which directly relates to ice rafted debris (IRD; Figure 4).

## 2.2 Abundance counts

The core sections of the entire working half were sampled every cm, resulting in 1 cm sample slices that were each washed over a 63 μm sieve mesh, dried overnight at ~75°C and subsequently size fractionated into 63-150 μm and > 150 μm. For abundance counts of planktonic foraminifera, slices every 4 cm were used, the counts were performed on *G. bulloides*; *N. pachyderma*; 'other foraminifera' and terrigenous grain in the >150 μm size fraction. The relative abundance was calculated from a sum total of both planktonic foraminifera (*G. bulloides*; *N. pachyderma*; and 'other foraminifera') and ice rafted debris (IRD: Stained-Quartz; Cloudy-Quartz; Bright-Quartz; Quartz; Sandstone, Igneous, Obsidian-glass; Rhyolitic-glass and '*Other*') counts and this does have complications for the relative abundance of foraminiferal species as it is a closed sum. Everything was counted and identified on a minimum of 200 grains after splitting with an OTTO-micro-splitter. The ratio of *N. pachyderma* to *G. bulloides* (Figure 4) is expressed as:

$$Ratio\ of\ NPS = \frac{N.pachyderma}{(N.pachyderma + G.bulloides)}, (1)$$

## 2.3 Single foraminifera stable isotope geochemistry ($\delta^{18}O$; $\delta^{13}C$)

For single shell stable isotope analysis, a continuous flow isotope ratio mass spectrometer was used (Feldmeijer et al., 2015; Metcalfe et al., 2015) based upon modifications (Breitenbach and Bernasconi, 2011) to the microvolume set-up (Spötl and Vennemann, 2003). Slices for isotope analysis were selected first at 10 cm resolution and then at specific sections down core every 2 cm. Two sections were analysed, a deglacial section between 300 cm and 420 cm and a glacial section between 515 and 565 cm. For each slice up to 20 shells of both left-coiling *N. pachyderma* and *G. bulloides* were picked at random from the 250 – 300 μm size fraction (Figure 5 to 6). No morphological differences were observed among the picked left coiling *N. pachyderma*. Each specimen was placed into a 4.5 ml exetainer vial, and the ambient air was replaced by He and subsequently digested in concentrated $H_3PO_4$ (45 °C for 160 minutes). The resultant $CO_2$-He gas mixture is transported to the GasBench II using a He flow through a flushing needle system where water is extracted from the gas using a Nafion tubing. The purified $CO_2$ is analysed in a Thermo Finnigan Delta$^+$ mass spectrometer after separation from other gases in a GC column. Isotope values are reported in the standard δ denotation with the ratio of heavy to light isotopes ($\delta^{18}O$) in per mil (‰) versus Vienna-Peedee Belemnite (V-PDB). The reproducibility of an international carbonate standard (IAEA-CO1) analysed is <0.12‰ (1 σ) for both $\delta^{18}O$ and $\delta^{13}C$, measured within the same run and at similar quantities (*i.e.*, producing similar amplitude on mass 44) to a single foraminifer. Based upon the amplitude of mass 44 which correlates with shell weight, foraminifera are estimated to weigh > 10 μg. Following Ganssen et al. (2011), data was screened for anonymous values, leading to outlier corrected values (red datapoints in Figure 5B, 5C, 6B and 6C) for both the deglacial section (*N. pachyderma*: $n_{total}$ = 414; $n_{outlier\ corrected}$ = 388; $n_{outlier}$ = 26; *G. bulloides*: $n_{total}$ = 439; $n_{outlier\ corrected}$ = 424; $n_{outlier}$ = 15) and glacial section (*N. pachyderma*: $n_{total}$ = 490; $n_{outlier\ corrected}$ = 474; $n_{outlier}$ = 16; *G. bulloides*: $n_{total}$ = 496; $n_{outlier\ corrected}$ = 474; $n_{outlier}$ = 22). Due to the nature of the mixture analysis (see section 2.5) the resultant outlier corrected populations differ from non-outlier corrected populations – although most of the uncorrected samples bearing outliers were beyond the models capacity to 'unmix' the results.

**2.4 Core stratigraphy and Age Model**

**2.4.1 Radiocarbon dating**

For radiocarbon dating of core T88-3P approximately 1 mg of pristine specimens of *G. bulloides* and *N. pachyderma* were picked from six samples of core T88-3P and analysed by Accelerated Mass Spectrometry (AMS) at the AMS laboratories

of Beta Analytic (Table 1; Figure 3). The open source MatCal (version 2.0) function for Mathworks MatLab® (Lougheed and Obrochta, 2016) was used to calibrate conventional radiocarbon age to a calendar age, using the Marine13 Calibration curve (Reimer et al., 2013) and a reservoir age of 400 $^{14}$C years with an error of 200 $^{14}$C years, expressed mathematically as ΔR: 0 ± 200 $^{14}$C yr (Reimer et al., 2013). The 95% confidence limits for the calendar age, in kyr BP, of each sample are given in Table 1.

**2.4.2 Age model construction**

Two age models were produced that used two independent methods, first a radiocarbon only age model and a second $\delta^{18}$O stratigraphy only age model (Figure 3). The first radiocarbon age model uses the 6 radiocarbon dates, using the maximum likely calendar age (in cal. yr. BP) from Table 1, the age model consisting of independent age markers places the deglacial period between ~410 and ~290 cm down core. A change in sedimentation occurs ~10,000 years ago at the core site, from

a slow glacial SAR (~10 cm/kyr) to a rapid interglacial SAR (~30 cm/kyr), this may reflect the sites location and the position of the polar front (Figure 1). The radiocarbon date at 500 cm is older than > 35 kyr, therefore the calibrated age can be considered less robust than the younger radiocarbon dates. A second, independent, down-core age model using $\delta^{18}$O stratigraphy was constructed as the deeper depths (> 500 cm) of the core are beyond the limits of radiocarbon dating. This stratigraphy utilised the cosmopolitan upper ocean dweller *G. glutinata* and the subpolar-temperate upper ocean dweller

*G. bulloides*, these species were measured for $\delta^{18}$O and $\delta^{13}$C using pooled specimens (2 groups of 5-10 specimens) picked from the 250 - 300 μm size fraction (Figure 2), which were placed in mono-specific groups within a 15 ml exetainer vial. Analyses followed the same procedures as for single shell analysis, but with considerably better reproducibility of international standards (1 σ <0.10‰) for the larger sample mass (~100 μg). The average $\delta^{18}$O of *G. bulloides* and *G. glutinata* was tuned to composite record of North Greenland Ice Core Project (NGRIP) (Rasmussen et al., 2008) on the

GICC05 timescale (subtracting 50 years to allow for a comparison between BP and b2K). Given the differences in resolution between a marine core and an ice core, the average $\delta^{18}$O of *G. bulloides* and *G. glutinata* was tuned to a filtered NGRIP signal: here we use the average of 'envelope' that reproduces the magnitude (highest value, lowest value) using a discrete Fourier transform with a Hilbert FIR filter of length 100. The two age models agree for the sections of the core between 0 and 30,000 years, however older than 35 kyr the two age models diverge however this may be simply a function

of the limitation of the radiocarbon calibration curve >35 kyr.

Using two independent age models, two estimates of the sedimentation rate have been made (Figure 3). The background sedimentation rate varies between 2.5 and 40 cm/kyr, being noticeably slower during glacial periods and much faster during the Holocene (Interglacial). During intervals of high ice rafted debris (IRD) input the sedimentation rate noticeably varies.

With respect to intervals chosen for single foraminiferal analysis, the SAR increases over the first interval (Green boxes in Figure 3) during the deglacial interval yet SAR stays relatively constant for the second interval chosen (Figure 3). Although given the depth in core, for this second interval only one estimate of SAR can be made.

## 2.5 Statistical analysis: End member modelling of SFA

Marine sediments reflect an averaged record over time, ranging from months to multiple centuries. However, if the individual components have distinct markers such as different $\delta^{18}O$ values, then the original distributions can be statistically unmixed into two or more univariate normally distributed populations using an un-mixing function (e.g. Hammer et al., 2001; Weltje, 1997; Weltje and Prins, 2003; Wit et al., 2013). Mixture analysis was carried out on outlier corrected samples (Figure 7) using the open source PAST (version 3.10) palaeontological statistics software (Dempster et al., 1977; Hammer et al., 2001). Using the end member modeling algorithm of Dempster et al. (1977), PAST estimates the mean, standard deviation and proportion of each population (see, Hammer et al. (2001) for a discussion of the assumptions of the mixing model). These solutions can be tested by two methods: the log likelihood value in which a 'better' result produces a less negative value, and a minimum in Akaike Information Criterion (AIC) value indicating that the chosen number of groups has a good fit without subsequent overfitting. An additional output of this mixture analysis is to assign each individual to the most probable population.

## 2.6 Modern Sediment trap record and Ocean reanalysis

As the modern analogue of our distinct isotopic 'end-members', we used seasonally resolved sediment trap time-series representing the modern polar, subpolar and temperate North Atlantic (Figure 1). Three such sediment trap records are available (Figure's 1 and 8) from (a) the polar Greenland-Norwegian Sea over the Iceland Plateau (IP – Wolfteich, 1994), (b) the subpolar Irminger Sea (IRM; Jonkers et al., 2010; Jonkers et al., 2013; Jonkers and Kučera 2015) and (c) the temperate mid North Atlantic (NABE48 from the North Atlantic Bloom Experiment; Wolfteich, 1994). Ocean reanalysis S4 (Balmaseda et al., 2013) was used to complete the temperature and salinity profiles associated with each sediment trap time-series, both with respect to time and depth (Figure 8). Ocean reanalysis data was converted from date into sediment trap cup number using a Mathworks MatLab® function: the monthly temperature and salinity data was first interpolated to one day resolution, using the interp1 function, the opening and closing dates of successive cups were then found, temperature and/or salinity presented in figures represent the opening and therefore the closing of the previous cup used. Since the IRM time-series represents several years we generate both a time averaged flux record as well as an average profile for both temperature and salinity. The time averaged flux is calculated by finding corresponding bimonthly (cup opening interval: 14 days) trap opening and closing days and averaging the resultant flux.

## 3. Results

### 3.1 IRD, Abundance

The upper ~290 cm of core T88-3P is Holocene in age as evidenced by near uniform values of pooled specimen $\delta^{18}O$ values, Log(Ca/Ti), IRD and the ratio NPS (Figure 4). Between 290 and 410 cm, the deglacial interval, the ratio NPS approaches 1.0, IRD 20% and a minimum Log(Ca/Ti) of 0.9. The minimum in Log(Ca/Ti) occurs prior to the increase in IRD though coeval with the increase in the ratio of NPS. Between 425 and 937 cm the ratio NPS and percentage of IRD appear to covary whilst the Log(Ca/Ti) shows an inverse, with a minimum in Log(Ca/Ti) during IRD events.

### 3.2 Single shell $\delta^{18}O$: *N. pachyderma*

Single shell analysis of *N. pachyderma* (Figure 5) was performed on a deglacial interval (300 to 420 cm) and a glacial interval (515 – 565 cm). Our glacial results show the abundance during this period has two peaks centred at 515 and 560

cm, with a large proportion of the data occurring within an interval of lower abundance. Single shell $\delta^{18}O$ average values and standard deviation lie between 2.5 and 4.5 ‰, with a remarkable consistent spread in the values, the lightest $\delta^{18}O$ values occur during lower species abundance. In comparison, our deglacial results show the abundance decreasing from a peak centred at 380 cm. Single shell values appear to get heavier from 420 to 360 cm, with a reduced spread, before becoming lighter and having a larger spread between 360 and 340 cm. The $\delta^{18}O$ average values and standard deviation during this interval range from 5.5 to 2.5 ‰.

### 3.2 Single shell $\delta^{18}O$: *G. bulloides*

Single shell analysis of *G. bulloides* was performed (Figure 6) on a deglacial interval (300 to 420 cm) and a glacial interval (515 – 565 cm). Our glacial results show the abundance during this period has a single peaks centred at 540 cm, with a large proportion of the data occurring within an interval of high abundance. Single shell $\delta^{18}O$ average values and standard deviation lie between 3.5 and 1.5 ‰, with a remarkable consistent spread in the values. In comparison, our deglacial results show the abundance decreasing from a peak centred at 420 cm, reaching a lower limit between 380 and 360 cm, before rising again. Single shell values appear to be relatively consistent from 420 until 360 cm, with a reduced spread, before becoming lighter and having a larger spread between 360 and 340 cm. The $\delta^{18}O$ average values and standard deviation during the deglacial interval range from 4 to 1.5 ‰.

### 3.5 Single shell $\delta^{18}O$ standard deviation

Comparison between the glacial and deglacial samples highlight the change in standard deviation between the two time periods: The glacial samples (515-565 cm) for *N. pachyderma* have a standard deviation corrected for outliers ($\mu = 0.44$; min = 0.28; max = 0.61; $\sigma = 0.09$; $n_{groups} = 26$; $n_{within\ group} = 474$) and uncorrected for outliers ($\mu = 0.55$; min = 0.28; max = 0.85; $\sigma = 0.15$; $n_{groups} = 26$; $n_{within\ group} = 490$) lower than the standard deviation of the deglacial samples (300-420cm) irrespective of whether the data was uncorrected ($\mu = 0.70$; min = 0.26; max = 1.52; $\sigma = 0.31$; $n_{groups} = 22$; $n_{within\ group} = 414$) or corrected for outliers ($\mu = 0.49$; min = 0.13; max = 1.11; $\sigma = 0.26$; $n_{groups} = 22$; $n_{within\ group} = 388$). The deglacial interval has a larger range in standard deviation that the glacial intervals, with both the smallest and largest spread as represented by the sample standard deviation. Whereas, the deglacial uncorrected ($\mu = 0.46$; min = 0.24; max = 0.72; $\sigma = 0.13$; $n_{groups} = 23$; $n_{within\ group} = 439$) and corrected ($\mu = 0.38$; min = 0.19; max = 0.65; $\sigma = 0.13$; $n_{groups} = 23$; $n_{within\ group} = 424$) data for *G. bulloides* is somewhat similar for the same species glacial uncorrected ($\mu = 0.69$; min = 0.33; max = 1.53; $\sigma = 0.28$; $n_{groups} = 26$; $n_{within\ group} = 496$) and corrected ($\mu = 0.48$; min = 0.24; max = 0.88; $\sigma = 0.16$; $n_{groups} = 26$; $n_{within\ group} = 474$) data.

### 3.5 Single shell $\delta^{18}O$ populations

Our results show that $\delta^{18}O$ values of both *G. bulloides* and *N. pachyderma* are predominately unimodally distributed during the last Glacial until about 21 ka BP (Figure 5 and 6). In the glacial section (515-565 cm) only a few samples appear to have a second population for *N. pachyderma* (5 out of 26 samples) and *G. bulloides* (9 out of 26 samples) with their appearance being more sporadic than systematic. This shows remarkably contrast with the deglacial section (300-420 cm). Whilst the distribution of *G. bulloides* $\delta^{18}O$ values remains predominately unimodal throughout the deglacial section (13 out of 23 samples), the $\delta^{18}O$ values of *N. pachyderma* develops striking bimodality (12 out of 22 samples; Figure's 5 and 7). For *N. pachyderma* the distributions can be statistically unmixed into two discrete populations (Hammer et al., 2001)

in varying numbers of specimens: one high in $\delta^{18}O$ persisting from the Glacial (population P1) and a second population low in $\delta^{18}O$ appearing at the onset of the deglaciation (population P2). The difference in $\delta^{18}O$ between population P1 and P2 amounts to $0.9 \pm 0.4$ ‰ and persists for (as estimated by the age models, Figure 3) about 10 ka while absolute values gradually decrease by 1.6‰ (Figure's 5 and 7). At the end of the last deglaciation, P1 disappears and the $\delta^{18}O$ values of *N. pachyderma* become once more unimodal, now for P2, shortly before disappearing entirely until the present day. Carbon isotope values ($\delta^{13}C$) measured on the same shells of *N. pachyderma* do not appear to show this bimodal distribution, precluding the possibility that the two populations in $\delta^{18}O$ represent a similar season but grew their shells at different depths given the enrichment and depletion with depth in seawater $^{13}C$ associated with phytoplankton growth and decay. Since the $\delta^{18}O$ values of *N. pachyderma* exhibit bimodality our findings down core could equate with seasonal gradients and species successions as observed in modern time-series from sediment traps deployed in the modern North Atlantic (Figure's 1 and 8) at 48°N (temperate), 59°N (subpolar) and 68°N (polar).

## 4. Discussion

### 4.1 Modern analogue

Modern conditions that mimic Glacial times down core are presently found in the polar Greenland-Norwegian Sea where productivity is limited by low light conditions, deep mixing and intermittent sea ice cover (Kučera et al., 2005). At 68°N, late summer insolation and thermal stratification spur a plankton bloom (August-September). At the same time planktonic foraminifera produce a single high maximum in the shell flux of *N. pachyderma* with few *G. bulloides* (Jonkers and Kučera, 2015) at temperatures of 3-5 °C, before the arrival of meltwater (Figure 8a). Further south, at 59°N in the subpolar Irminger Sea, the flux of *N. pachyderma* is bimodal, with an early 'cold' population being produced in April-May (4-6 °C) and a late 'warm' population occurring in August-September (7-9 °C) that are separated by a single pulse in *G. bulloides* (Jonkers et al., 2010; Jonkers et al., 2013) (Figure 8b). Neither of the *N. pachyderma* populations from IRM display significant morphological differences (Jonkers et al., 2010; Jonkers et al., 2013). By contrast, modern shell fluxes in the temperate North Atlantic at 48°N, close to our core site, are dominated by *G. bulloides* in early summer yet completely devoid of *N. pachyderma* year around (Wolfteich, 1994) (Figure 8c).

Spatial differences in modern seasonality observed in the polar to temperate North Atlantic provide modern analogues for interpreting temporal changes in the sediment record in terms of the seasonal modes developing since the last Glacial. During peak glacial times the northern North Atlantic is covered by sea ice down to 45°N (Kučera et al., 2005) (Figure 1) except for a short interval in late summer allowing for a period of high productivity dominated by *N. pachyderma* (P1) as seen in the modern Norwegian-Greenland Sea at 68°N (Jonkers and Kučera, 2015) (Figure 8a). With the reduction in (sea-) ice cover during the initial deglaciation *N. pachyderma* starts occurring earlier in summer, persisting at the same low temperatures. As the deglaciation progresses the 'cold' population (P1), with a similar unimodal distribution as in the Glacial, is joined by a second 'warm' population (P2) that starts appearing in late summer. The isotopic difference between P1 and P2 ($0.9 \pm 0.4$ ‰) corresponds to a temperature offset of about ~4 °C, the same as observed today at 59 °C (Jonkers et al., 2013).

The modern seasonal succession of P1 and P2 generates the same bimodality we observe in the $\delta^{18}O$ of the mixed *N. pachyderma* populations during the deglaciation in our core record (Figure 7). Such bimodality may well be an expression

of two genetically different but morphologically identical "cryptic species" among *N. pachyderma* (Bauch et al., 2003; Darling et al., 2000; Kučera and Darling, 2002). Indeed, morphologies are indistinguishable among our encrusted specimens from the 250-300 μm both in our cored sediment and in modern *N. pachyderma* from the time-series sediment traps at 59°N during both seasonal maxima, regardless of the size fractions used (Jonkers et al., 2010; Jonkers et al., 2013).

Unfortunately, the lack of organic matter, due to the process of low temperature ashing, concentrate and isolate the mineral shells from organic matter leaving a clean residue for isotope and chemical analysis, limits the ability to genetically analysis trap specimens. At our core site, increasing temperatures would have first caused the disappearance of the "cold" water population P1 (~9.5 ka BP) followed shortly after by the disappearance of "warm" population P2 (Figure 7) when Holocene temperatures at this latitude exceed *N. pachyderma*'s upper tolerance limit of ca. 10 °C (Darling et al., 2006).

**4.2 Alternative mechanisms and scenarios**

Single specimen isotope analysis permits unravelling of mixed sedimentary assemblages into their constituent components. Here we show that the warming trend within the average $\delta^{18}O$ of pooled *N. pachyderma* is directly caused by the emergence of a "warm" population (P2) shifting the mean isotopic value toward a warmer signal, concealing the continued existence of the original "cold" (P1) population. Within the northern North Atlantic an abrupt change occurs from a single peak in

production during the LGM to two populations that remain approximately 4°C apart throughout the deglaciation, inferring that the difference in $\delta^{18}O$ is temperature driven, consistent with present day observations from subpolar sediment trap time-series. However, alternative scenarios that give the same or a similar solution for the existence of two populations can be envisaged. Below, we discuss other causal mechanisms that might be inferred from the data, including a low salinity meltwater effect (Duplessy et al., 1991), bioturbation (Lougheed et al., 2018) and/or population dynamics (Mix, 1987;

Roche et al., 2018).

**4.2.1 Warming trend or Meltwater pulse?**

Reconstructions of the $\Delta\delta^{18}O_{sw}$ anomaly between the LGM and Modern (Duplessy et al., 1991) suggest a series of regions above the southerly displaced Polar Front where freshwater and meltwater entered the North Atlantic in sufficient volumes to perturb the system, from continental ice meltwater and/or riverine input. Throughout the deglacial period, advances in

the subtropical water masses and retreats of the Polar Front occurred. Repeated invasion of high temperature and salinity waters into the Nordic Seas have shown that the deglacial period was inherently highly dynamic and thus unstable compared to the LGM as evidenced by isotopic (Duplessy et al., 1992; Kroon et al., 1997) and radiocarbon (Waelbroeck et al., 2001) measurements. Meltwater released into the northern North Atlantic during this time would have led to an increase in stratification and thus a decrease in SST altering the E-P balance that drives the poleward advection of

subtropical water high in both temperature and salinity (Duplessy et al., 1992). The two populations found in our core during the deglaciation might have resulted from one seasonal population experiencing meltwater and a second seasonal population occurring before or after a meltwater event. The presence of continental ice-rafted debris (IRD) down core in T88-3P, without a clear concomitant 'spike' in the $\delta^{18}O$, referred to in the literature as a 'meltwater spike' (Berger et al., 1977; Jones and Ruddiman, 1982) of either *G. bulloides* (Figure 2 and 6) or *N. pachyderma* (Figure 5) would suggest that

the difference in $\delta^{18}O$ between the two populations is dominated by temperature, consistent with previous studies showing no meltwater spike (Duplessy et al., 1996; Straub et al., 2013). Indeed, the presence of both foraminifera and IRD together down core does not necessarily imply cohabitation of the same environment, as the modern seasonal maximum in polar shell productivity occurs prior to the arrival of melt water from ice bergs (Figure 8). The extremely low values of

continental ice ($\delta^{18}$O: -30 to -40 ‰) should lead to $\delta^{18}$O and salinity anomalies in surface waters, but sea-ice formed from ocean water will have little impact on $\delta^{18}$O despite an impact upon salinity. Therefore, a concordial meltwater $\delta^{18}$O signal and the presence of IRD is not compulsory (Duplessy et al., 1996) with increased sea-ice formation predicted to occur during periods of increased freshwater and extended Arctic Ocean area (Duplessy et al., 1996).

### 4.2.2 Spatial rather than temporal populations: Shallow or deep?

Differences in depth habitat rather than timing might account for our observations. Depending on the structure of the water column, i.e. the depth of the surface mixed layer and the degree of stratification (see Metcalfe et al. (2015) for a discussion), the populations could represent one shallower and one deeper population that are not divided temporally but vertically within the water column (Figure 8). Observations from the subpolar IRM time-series sediment traps show that the first maximum occurs at an earlier time when the water column is well mixed, so that two vertically divided populations, i.e. one shallow and one deep would have a similar $\delta^{18}$O signature. The second maximum in IRM occurs at a later time, during increased water column stratification, therefore a shallow and deep population's $\delta^{18}$O should diverge. Therefore, only when the water column is stratified would it be possible to produce two theoretical populations different in $\delta^{18}$O, in much the same way as discrete species calcifying at different depths acquire an isotopic offset, enabling the use of $\Delta\delta^{18}$O as a proxy for past ocean stratification (Emiliani, 1954; Lototskaya and Ganssen, 1999; Mulitza et al., 1997). Following this line of reasoning, our results would suggest that the water column was more stratified during the deglaciation and well-mixed during the LGM and Holocene.

One approach to further differentiate between depths is the carbon isotope ($\delta^{13}$C) signal, as seawater $\delta^{13}$C has a distinct signature, due in part to photosynthetic fractionation in the surface ocean enriching the euphotic zone in $^{13}$C, exported organic matter may become remineralised at the base of the deep chlorophyll maximum enriching the euphotic zone in $^{12}$C at greater depth. Thus, the $\delta^{13}$C of foraminifera that have grown at different depths in these water masses should also have different values for each subpopulation, notwithstanding species specific vital effects. However, differences or similarities in carbon isotopes can also arise by alteration in food (either through grazing on different trophic levels and/or the types of food), if two seasonal populations of foraminifera existed either their food source thrived for longer or a succession of more oligotrophic tolerant phytoplankton occurred. Therefore, the carbon isotope signature can be related to both scenarios. By contrast, our results show no differences in $\delta^{13}$C signature between the two populations of *N. pachyderma*, notwithstanding inter-specimen variance. What is directly observable however, is that the IRM shows that there are two populations occurring seasonally. Second, there are no morphological differences observed between the specimens of *N. pachyderma* that isotopically belong to different populations in our core record, nor between the early and late summer maxima in *N. pachyderma* with a similarly distinct isotope composition in the modern sediment trap time-series. Most species undergo wall thickening with depth (Brummer et al., 1987, 1986; Hemleben et al., 1985; Reynolds et al., 2018; Steinhardt et al., 2015) whilst some, including *N. pachyderma* add a thick calcite crust with a different $\delta^{18}$O signature overprinting previous layers of the shell (Kozdon et al., 2009). This crust however is an ontogenetic feature (Brummer et al., 1987, 1986; Steinhardt et al., 2015) that is present in both seasons at IRM (Jonkers et al., 2010; Jonkers et al., 2013).

### 4.2.3 Sedimentary processes: Dissolution and Bioturbation

Seafloor processes such as dissolution and bioturbation may alter sediment populations in both isotope composition (Bard et al., 1987; Lougheed et al., 2017; Wit et al., 2013) and faunal composition (Bard, 2001; Löwemark, 2007; Löwemark et

al., 2008). Dissolution not only removes 'time' from the sediment but also leads to specimens being found together that have once been separated by centimetres of sedimentary material, as younger shells are deposited next to freshly exposed older shells (Lougheed et al., 2018). Similarly, depending on the oxygen content of sediments and the type and abundance of bottom fauna, bioturbation by benthic organisms can alter the sequence of cause and causality (Lougheed et al., 2018). Particle grain size distributions may also change due to bioturbation (Bard, 2001) if two species have differences in their absolute size, such as those measured here, it may show distinct isotope differences given species-specific size distributions (Brummer et al., 1986; Peeters et al., 1999).Thus, the two populations found in *N. pachyderma* $\delta^{18}$O could reflect relict specimens displaced in core-depth, and therefore in time, given there is a shift in the sedimentation rate of core T88-3P between 290 cm and 410 cm and not one between 515 and 565 cm (Figure 3). However, such sorting effects can be excluded here since both *N. pachyderma* populations and *G. bulloides* come from the same > 250 µm size fraction. It is important to note that for several depths in core this second population may only represent a few specimens ($n_{< 3\ specimens}$ = 3; and $n_{< 4\ specimens}$ = 5). Similarly, Löwemark et al. (2007; 2008) have shown that it is possible to have apparent differences due to the original abundance of the bioturbated species (Bard et al., 1987; Löwemark and Grootes, 2004). The difference between the two populations in *G. bulloides* and *N. pachyderma* could reflect a change in dominance of the foraminiferal assemblage, with bioturbation becoming more obvious in *N. pachyderma* as the species abundance reduces (Figure's 4 and 5).

Bioturbation is more easily detected during a period of pronounced climatic change, i.e., when the two end-member samples have the largest difference, as the signal of bioturbation becomes more pronounced. This may explain the difference in populations between the results from the deglacial (290 to 410 cm; Figure 7) and glacial (515 to 565 cm; Figure 7) sections despite both sections having similar abundance shifts (Figure's 4 to 6). As the resultant single specimen $\delta^{18}$O distribution is a product of species-specific temperature tolerances (Mix, 1987; Roche et al., 2018), the visibility of bioturbation is especially enhanced at periods of sharp climatic transition. If the climatic signal crosses through a species temperature tolerance then two separate warm and cold populations should exist separated both in time and core depth, bioturbation will then mix these populations together. However, we exclude this particular scenario because sedimentary features (Figure 4) indicate a lack of discernible mixing, i.e. the sharpness of the IRD percentage and the all indicate that bioturbation is at a minimum.

## 4.3 Palaeoceanographic implications: Probability of drawing from either population I or II

The implications for the climate of the past are twofold. First, our results suggest that there is more than one population of the polar *N. pachyderma* during the deglaciation and that its continued presence throughout much of this time period puts doubt to two discrete modes. The presence of both a colder population and warmer population suggests that this period is characterised by heightened seasonality, given that the climate conditions prevalent at ~56°N supported two populations of *N. pachyderma*. This heightened 'seasonality' is also visible in the increased standard deviation (Figure 5) for these samples. The second implication is that this causal mechanism (*i.e.*, seasonally distinct populations occurring during a climate transition) may not be captured using a pooled sample approach, given two distinct reactions to the same climate transition. It is important to note that *G. bulloides* also on occasion has more than a single population at this core site, however the species cosmopolitan and optimistic nature make it less surprising that expansion of seasonal variables that intersects the species tolerances will lead to an expansion of its ecological range (*e.g.*, Metcalfe et al., 2019). The appearance of a second population in *N. pachyderma* is more surprising because ecological expansion for this species can

only be unidirectional (i.e., into water masses with higher temperatures) given its dominance of the polar environment. Therefore, given the use of *N. pachyderma* as a polar water mass indicator, we chose to investigate how multiple populations would impact pooled analysis. The un-mixing algorithm used in this paper gives the probability of each distinct population, using each population and their calculated mean and standard deviation to generate a normal distribution for

the populations determined via statistical un-mixing (Hammer et al., 2001; Wit et al., 2013). Using this data it is possible to model the theoretical effect of sample size upon the resultant stable isotope measurements (Morard et al., 2016). For simplicity we assume, that each specimen contributes an equal weighting to the overall pooled stable isotope value, of course in reality each specimen will contribute an amount of $CO_2$ equal to its weight. This assumption will result in some error associated with our prediction of pooled specimen $\delta^{18}O$ values due to fractionation during conversion from $CaCO_3$ to

$CO_2$ (and $H_2O$). The theoretical specimens were picked from either population, or for those samples in which only a single population exists (at either limits of our sampling), using the rand function of MatLab. The function rand is statistically uniform throughout the range 0 to 1 and therefore can be used to construct a random number generator to define which population each theoretical specimen would have belonged to, using the following equations:

$$r = rand(N_{pool}, 1) \geq p\left(\delta^{18}O_{\text{pop. II}}\right), (2)$$

$$R1 = normrnd\left(\delta^{18}O_{\mu \text{ pop. I}}, \delta^{18}O_{\sigma \text{ pop. I}}, N_{pool}, 1\ \right), (3)$$

$$R2 = normrnd\left(\delta^{18}O_{\mu \text{ pop. II}}, \delta^{18}O_{\sigma \text{ pop. II}}, N_{pool}, 1\ \right), (4)$$

$$S = (1-r). * R1 + r. * R2, (5)$$

The number of pooled specimens (in-group analysis; $N_{pool}$) were varied between iterations of the model, so that 5, 6, 7, 8, 9, 10, 20, 30, 40, 50, 60, 70, 80, 90 and 100 draws/specimens were used for each subsequent iteration. For each depth

10,000 redraws were performed, because we use a large number for resampling (N = 10,000), and due to the central limit theorem, the average $\delta^{18}O$ between the different iterations (variable number of pooled specimens) remains near constant. Therefore, the 2.5[th], 25[th], 75[th] and 97.5[th] quantiles were used to visually compare the spread of the data between different numbers of pooled specimens (Supplementary Figure 2), and the probability of $\delta^{18}O$ values occurring calculated (Figure 9). The results of the model indicate that for a foraminiferal fossil population composed of more than one discrete

subpopulation, caution should be applied when using a small number of specimens for pooled analysis to ascertain an average state of the climate. Whilst the spread in values is narrower for some intervals; several down core samples have a spread of 1 to 1.5 ‰ (Figure 9).

The fact that a sample may have a single population for one or both species and intermittently two or more populations

(i.e., *N. pachyderma*) may further complicates species comparison (e.g. $\Delta\delta^{18}O$). The emergence of a second population within *N. pachyderma* during the last deglaciation at the species southerly boundary, indicates that other species with multiple abundance or size maxima (Schmidt et al., 2004a; Schmidt et al., 2004b) may have a similarly hidden seasonal complexity within the stable isotope composition of pooled specimens. If these populations do not represent ecophenotypes, but instead are analogous to cryptic speciation in which populations are indistinguishable morphologically

(Kučera and Darling, 2002; Morard et al., 2016), then pooled isotope measurement of such a sample will accidentally 'pick' from multiple populations. Therefore, the wide use of *N. pachyderma* isotopes as a measure of sea-level rise, rate of deglaciation or ice volume change based upon the $\delta^{18}O$ of pooled specimens may be unduly skewed.

## 5. Conclusions

Our findings expose and resolve the seasonal complexity that exists hidden in the $\delta^{18}O$ produced within pooled specimens whilst highlighting the usefulness of integrating down core studies with modern time-series observation in the interpretation of species ecology for palaeoceanographic research. Using sediment trap time-series data as modern keys to past climate conditions our results imply that conditions existing today within the subpolar Irminger Sea prevailed at significantly more southerly latitudes throughout the last deglaciation. The remarkable difference between the transition (Deglaciation) and the two climatic modes (Glacial and Interglacial), suggests that the mid North Atlantic has an intermediate "deglacial" stable mode, rather than gradually shifting from Glacial to Interglacial. Our observation of a distinct bimodality throughout the deglaciation has important implications for how $\delta^{18}O$ records can be interpreted given present-day seasonality, however the interpretation of *N. pachyderma* as two populations instead of one is consistent with previous studies. Therefore, the common use of this species as a measure of sea-level rise, rate of deglaciation or ice volume change, or ocean warming and stratification based upon the $\delta^{18}O$ of pooled specimens may be unduly skewed.

**Data availability**

Upon publication the data of APNAP II T88-3P will be uploaded to a data repository.

**Sample availability**

Access to APNAP II T88-3P material should be done via request to Gerald Ganssen (VUA).

**Author Contributions**

G.M.G. was Chief Scientist of APNAP II (RV *Tyro*) during core retrieval, initiated and supervised the study, together with G.-J.B. W.F. conducted the investigation. J.v.'t. H. performed abundance counts. G.-J.B, W.F., B.M. and G.M.G. contributed to data analysis and interpretation. B.M. made the figures and performed statistics. G.-J.B. and B.M. wrote the manuscript with contributions from all authors.

**Competing Interests**

The authors declare no competing interests.

**Acknowledgements**

The captain and crew of the RV *Tyro* are thanked for core retrieval during APNAP II (1988). Core scanning was supported by the Netherlands Organization for Scientific Research (NWO) through the SCAN2 program on advanced instrumentation. Rineke Gieles is thanked for assisting with XRF core scanning. Sample treatment and preparation was conducted in the Sediment Laboratory of the Vrije Universiteit Amsterdam (VUA). Hubert Vonhof and Suzanne J. A. Verdegaal–Warmerdam (Stable isotope laboratory of the VUA) are thanked for assistance with stable isotope analysis. This is a contribution to the Darwin Center for Biogeosciences project "Sensing Seasonality" and the NWO funded project

"Digging for density" (NWO/822.01.0.19). BM was supported by a Laboratoire d'excellence (LabEx) of the Institut Pierre-Simon Laplace (Labex L-IPSL), funded by the French Agence Nationale de la Recherche (grant no. ANR-10-LABX-0018). The authors wish to thank the Integrated Climate Data Center (ICDC, DE) for their online live access servers that provided access to atlas and reanalysis data used within this study.

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

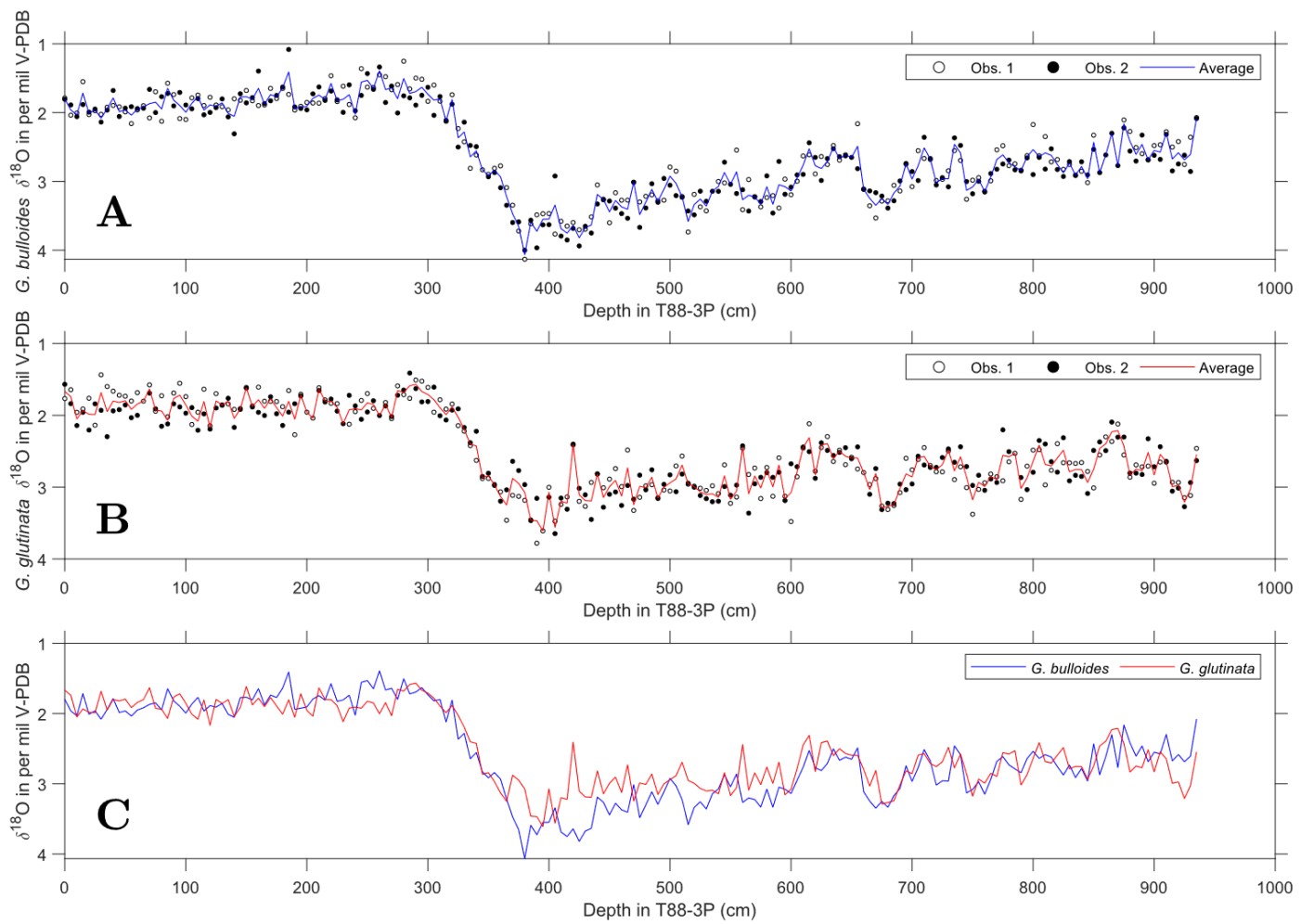

**Figure 2. Raw pooled foraminifera stable oxygen isotope data and computed averages. (A) Pooled stable oxygen isotope data of** *G. bulloides* **and (B)** *G. glutinata***, for each sample two groups (white and black dots) were measured and the (C) computed average for both species (lines in A to C).**

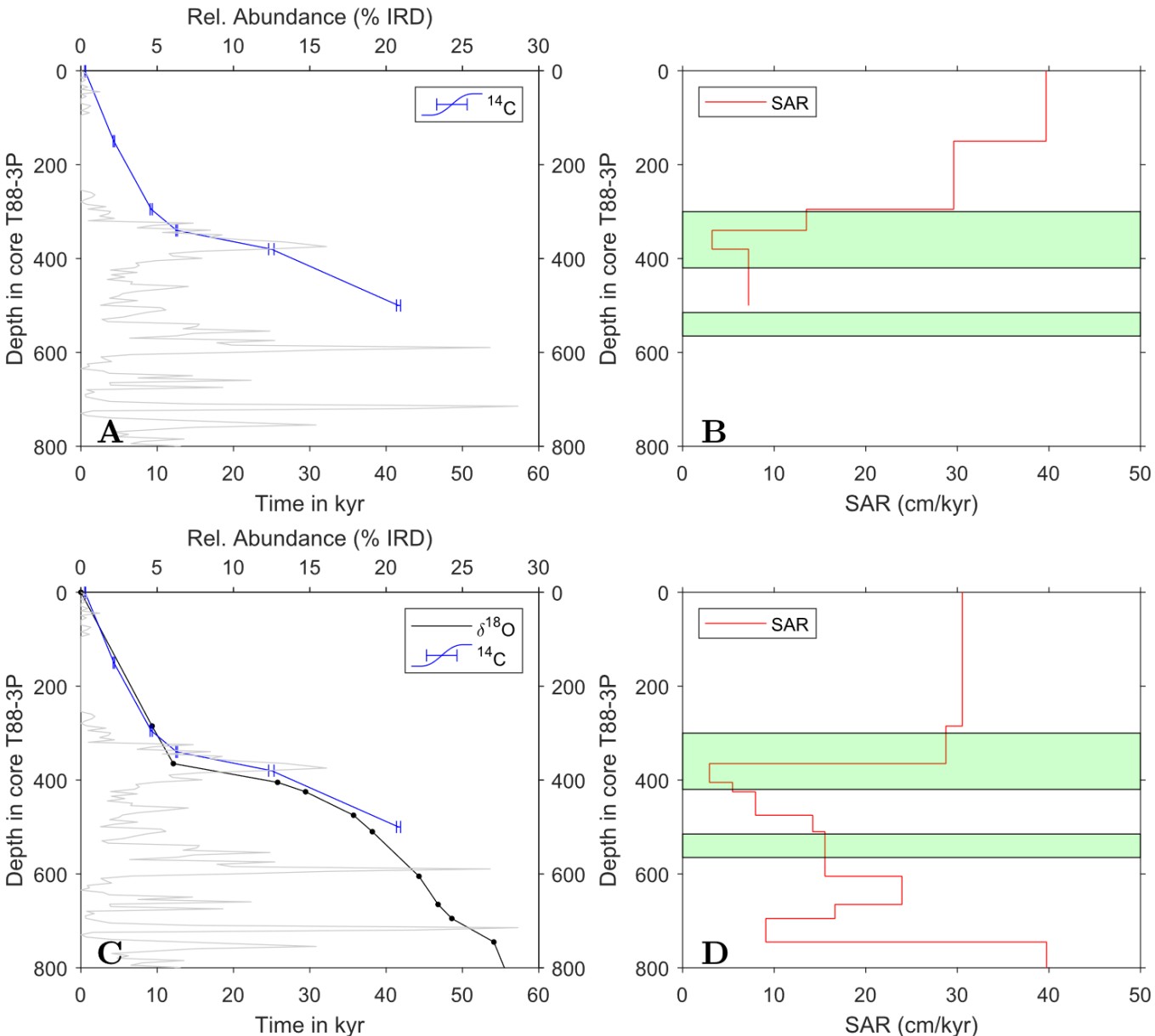

**Figure 3. Age models of core T88-3P and their respective SAR. (A)** Radiocarbon age model (see Table 1 and Supplementary Table 1 for radiocarbon ages) and **(B)** the estimated sediment accumulation rate (SAR). **(C)** Oxygen isotope stratigraphy of pooled measurements of *G. glutinata* and *G. bulloides* (see, Figure 2) tuned to NGRIP (see Supplementary Table 2 for tie-point estimates). For comparison, radiocarbon age model (blue line) is plotted alongside the tuned oxygen isotope age model (black line). Ice rafted debris (IRD) down core is plotted (grey) alongside the age models. **(D)** The estimated SAR for the tuned oxygen isotope age model. Green panels in **(B)** and **(D)** highlight depths in core where single foraminifera stable isotope analysis was performed. See, Table 1 and Supplementary Table 1 for Radiocarbon Measurements, and Supplementary Table 2 for δ¹⁸O tie-points.

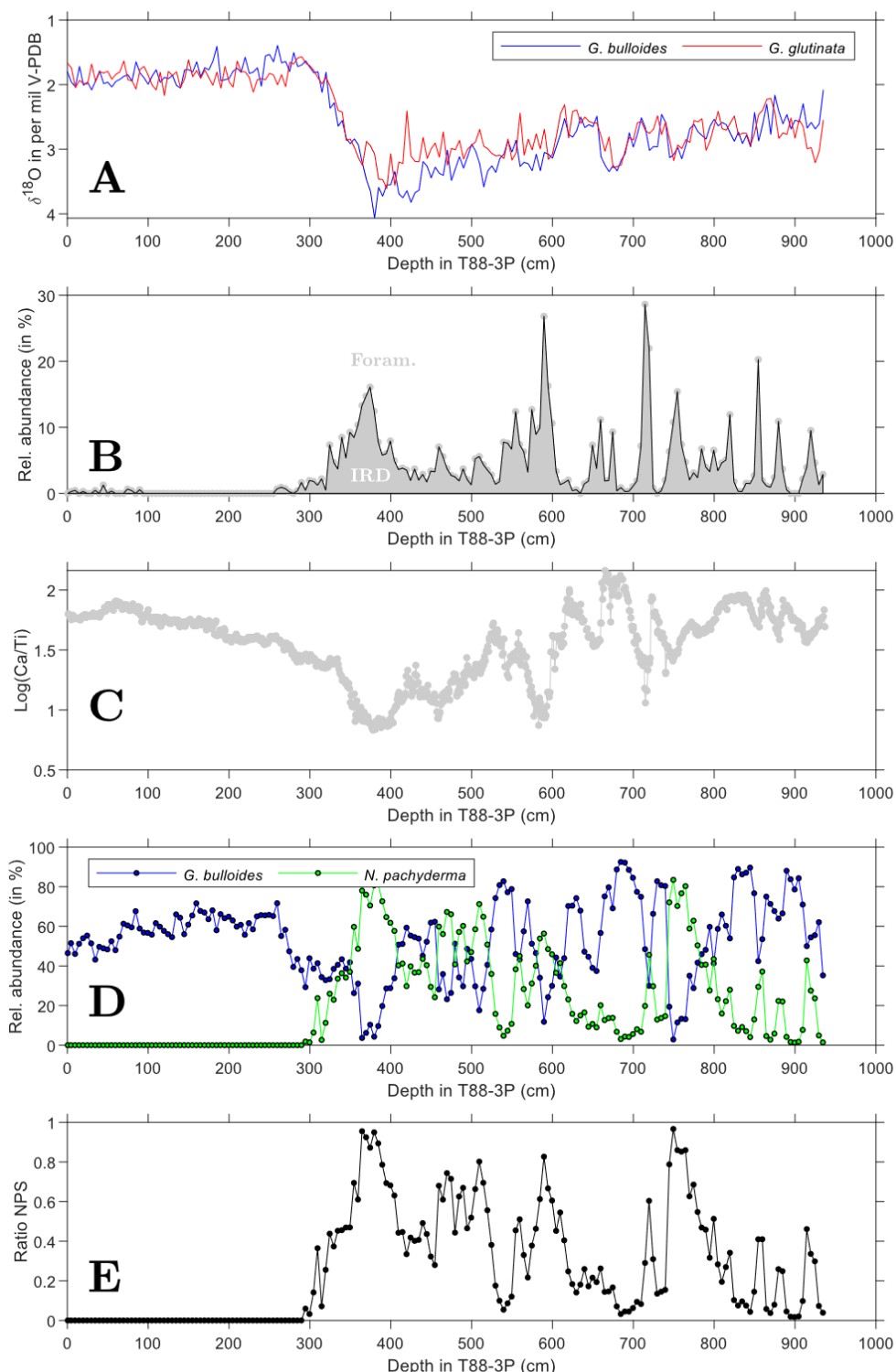

**Figure 4. Sediment and abundance data against depth in core. (A) Average oxygen stable isotopes of *G. bulloides* (blue) and *G. glutinata* (red), from Figure 2. (B) Relative abundance of ice rafted debris (IRD), calculated as the amount of IRD relative to both foraminifera and IRD from approximately 200 particles. Grey area reflects the relative abundance of IRD whereas, white area reflects the relative abundance of foraminifera. (C) The logarithmic ratio of Ca and Ti (Log(Ca/Ti)) counts per second (CPS) as measured by X-Ray Fluorescence (XRF) core scanning. (D) The relative abundance of the planktonic foraminifera species *G. bulloides* (blue) and *N. pachyderma* (green), the relative abundance is based upon the counts of IRD and foraminifera (see panel B). Depths reflect the mid-point of the sample. (E) Ratio of the relative abundance of *N. pachyderma* and *G. bulloides* (see equation 1).**

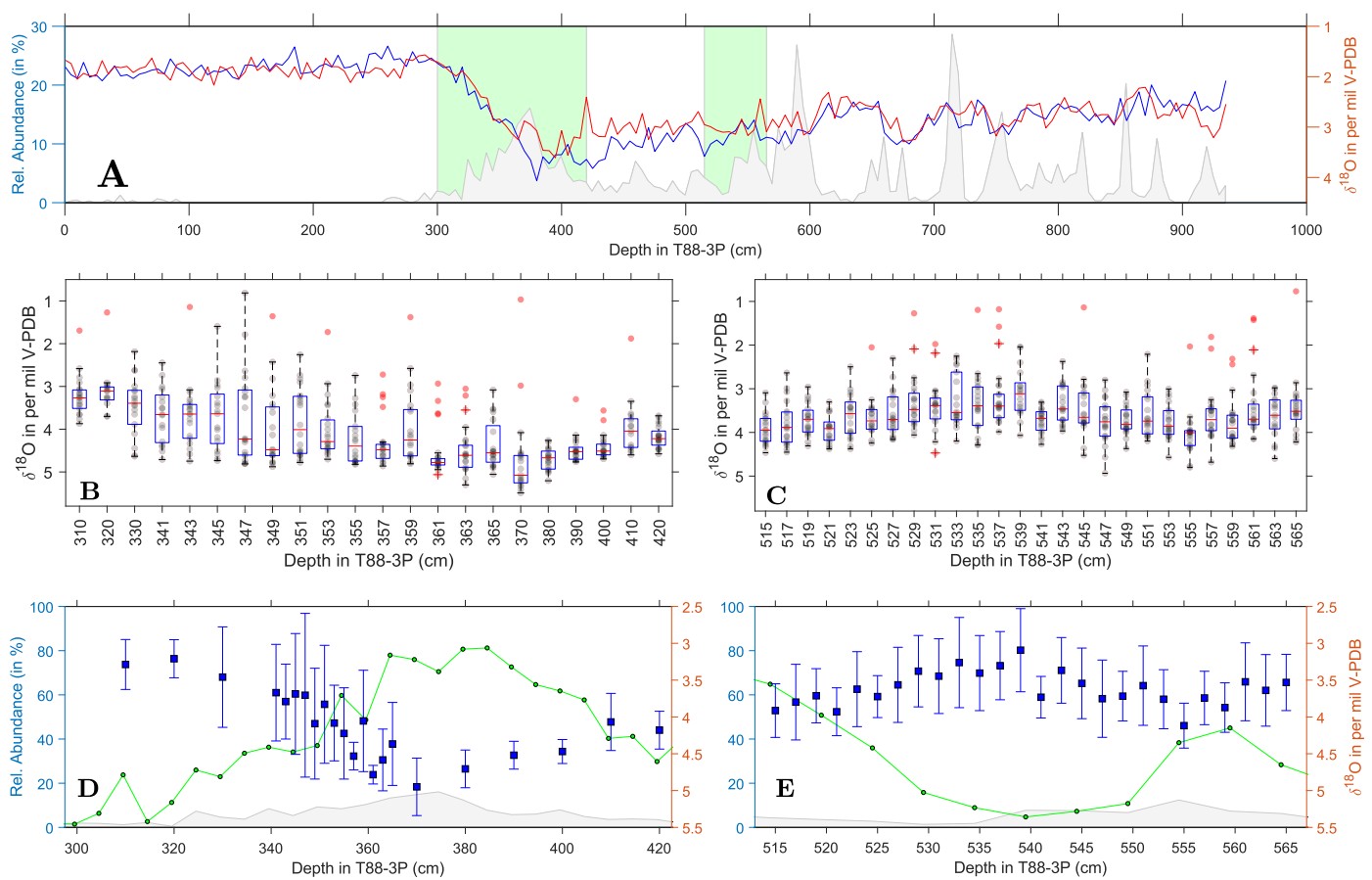

**Figure 5. Single foraminifera stable isotope data:** *N. pachyderma*. **(A)** Pooled average oxygen stable isotopes of *G. bulloides* (blue) and *G. glutinata* (red), from Figure 2, and Ice Rafted Debris (gray) for the same samples (Figure 3 and 4). Green areas represent the deglacial (B and D) and glacial (C and E) samples. **(B – C)** Raw stable isotope values (grey) of *N. pachyderma* for each sample and the samples respective outliers (red). Overlain are the statistical features of the distribution outlined by a box and whisker plot (median, upper and lower quartile). **(D – E)** Average values and the standard deviation (blue) of the outlier corrected oxygen isotope data plotted alongside the abundance of *N. pachyderma* (Green; see Figure 4) and IRD (Grey; see Figure 4).

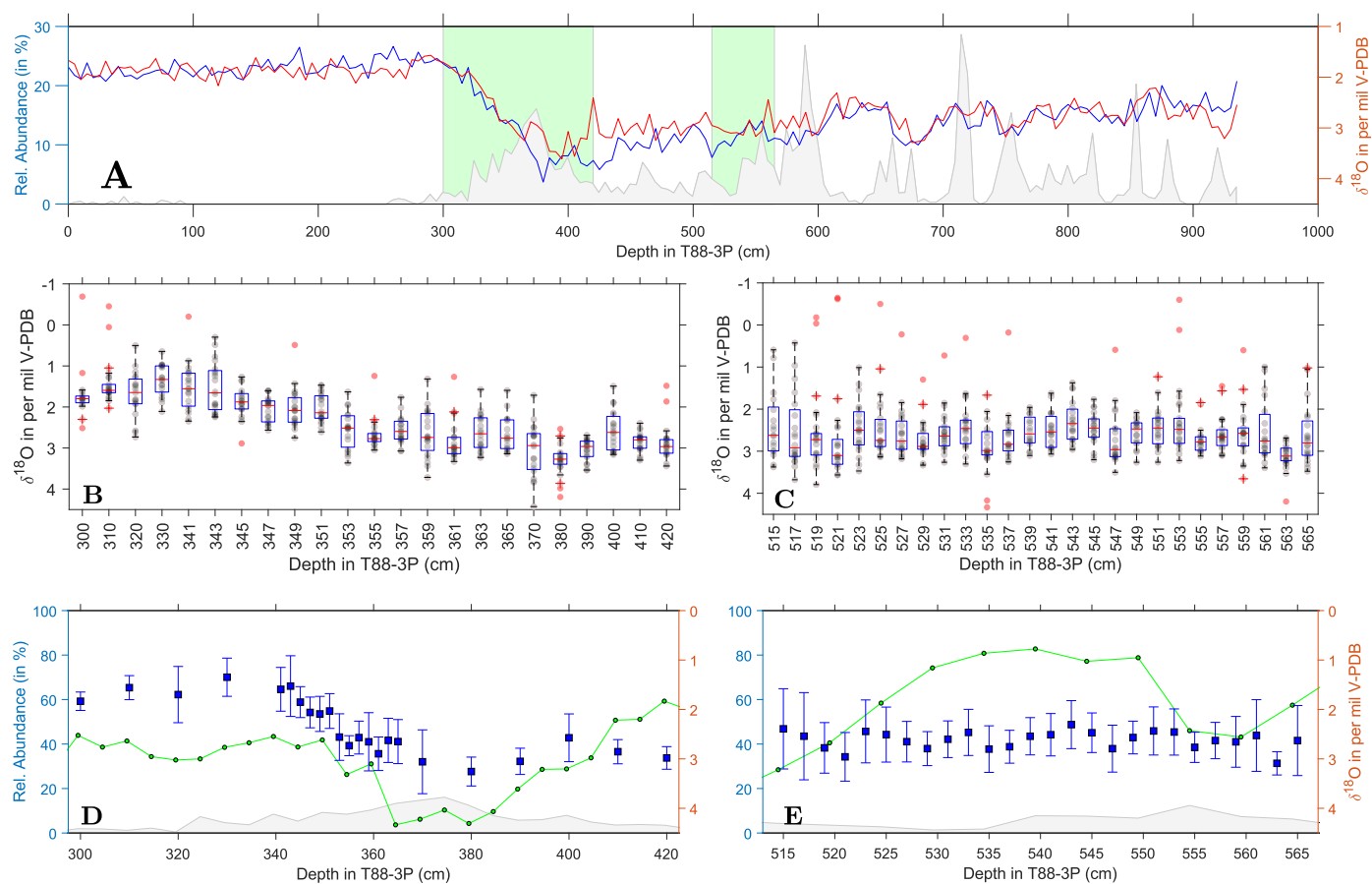

**Figure 6. Single foraminifera stable isotope data:** *G. bulloides*. **(A)** Pooled average oxygen stable isotopes of *G. bulloides* (blue) and *G. glutinata* (red), from Figure 2, and Ice Rafted Debris (gray) for the same samples (Figure 3 and 4). Green areas represent the deglacial (B and D) and glacial (C and E) samples. **(B – C)** Raw stable isotope values (grey) of *G. bulloides* for each sample and the samples respective outliers (red). Overlain are the statistical features of the distribution outlined by a box and whisker plot (median, upper and lower quartile). **(D – E)** Average values and the standard deviation (blue) of the outlier corrected oxygen isotope data plotted alongside the abundance of *G. bulloides* (Green; see Figure 4) and IRD (Grey; see Figure 4).

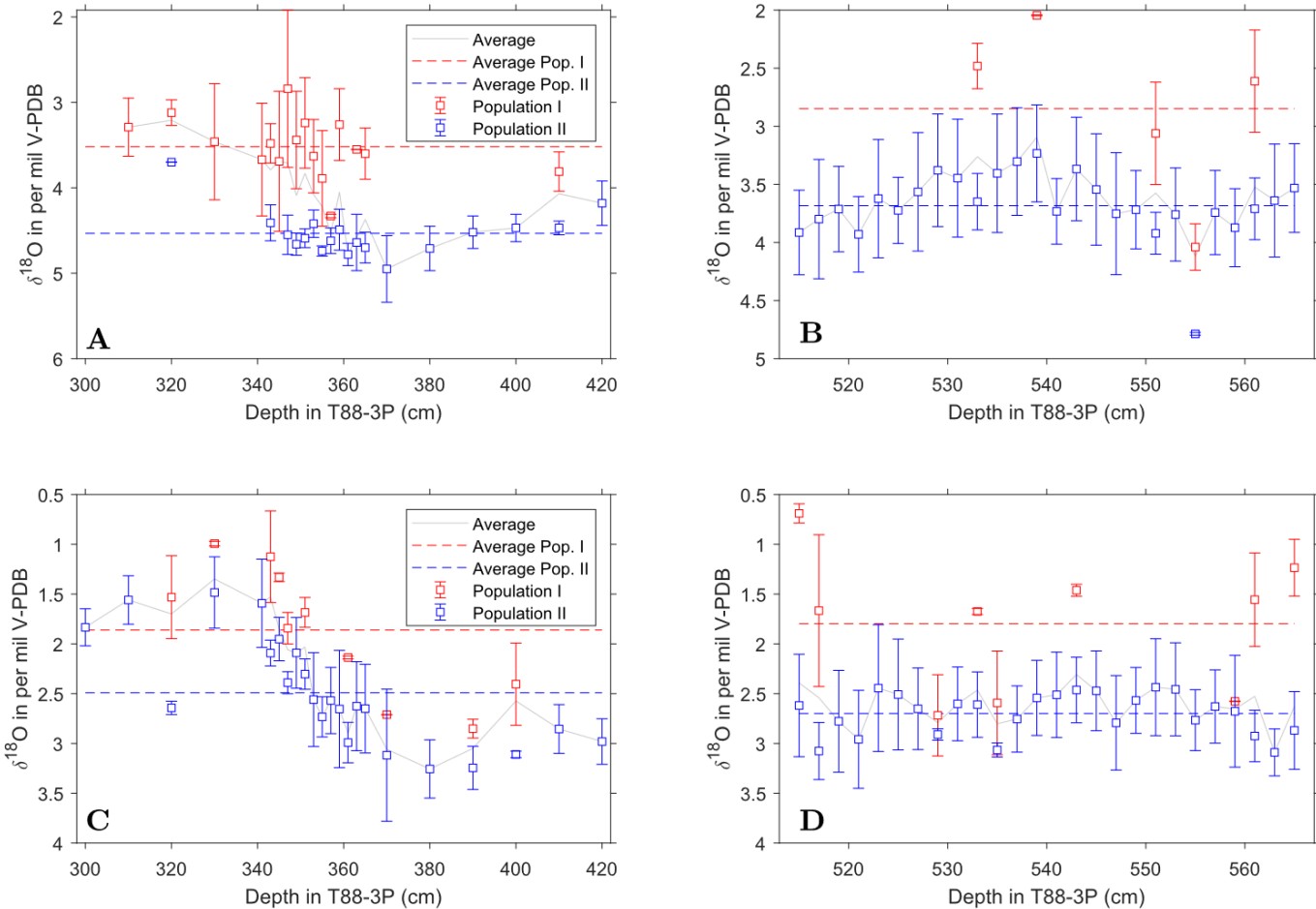

**Figure 7: Average δ¹⁸O value of single shell populations for specimens of *N. pachyderma* and *G. bulloides* across the deglaciation and for the glacial interval. (A and B) Mean and standard deviation of distinct populations of *N. pachyderma* plotted against core depth. (C and D) Mean and standard deviation of distinct populations of *G. bulloides* plotted against core depth. Calculated values for Population I and II, as determined from mixture analysis (Hammer et al., 2001). Vertical bars represent the standard deviation for each population, depths where multiple symbols are present are where it is not possible to distinguish statistically either one or more populations, these thus represent a single population of the sample to the left. Horizontal dashed lines represent the averages for population I and II, grey line is the total population average as would be reconstructed from pooled shell analysis. Above 300 cm (< ~10 kya), the Holocene, *N. pachyderma* has disappeared from the site**

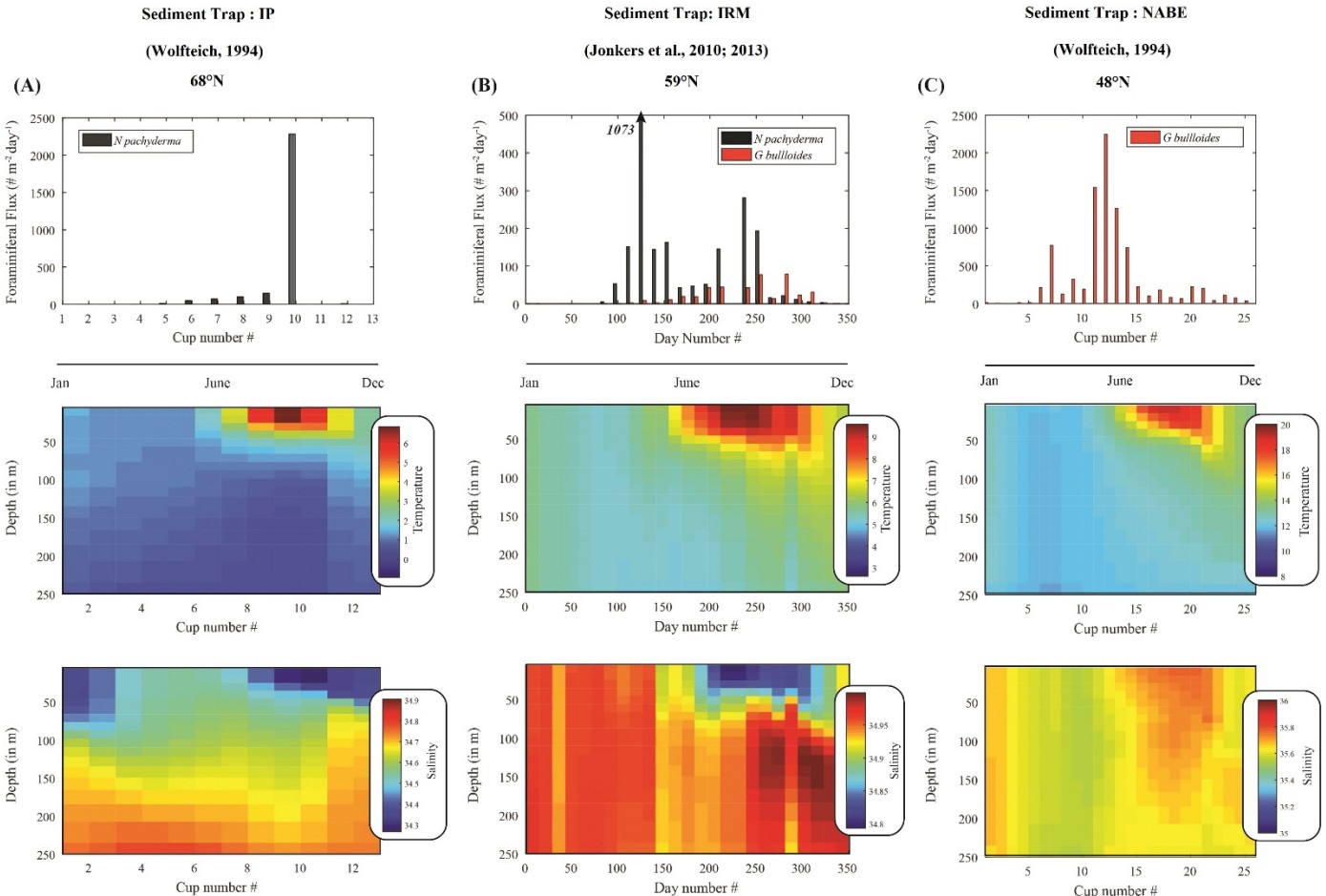

**Figure 8: Seasonal succession in the modern North Atlantic. Top panel, fluxes of *N. pachyderma* (grey) and *G. bulloides* (blue) from sediment traps in (A) the polar Greenland-Norwegian Sea (Wolfteich, 1994), (B) subpolar Irminger Sea (Jonkers et al., 2010; Jonkers et al., 2013; Jonkers and Kučera, 2015) and (C) temperate mid North Atlantic (Wolfteich, 1994), same labels as in Fig. 1. Middle and Bottom panels represent the temperature and salinity from ocean reanalysis ORAS S4 (Balmaseda et al., 2013). For (A) and (C) cups have been rearranged to progress from January – December. The fluxes, temperature and salinity given represent an averaging over the trap deployment period.**

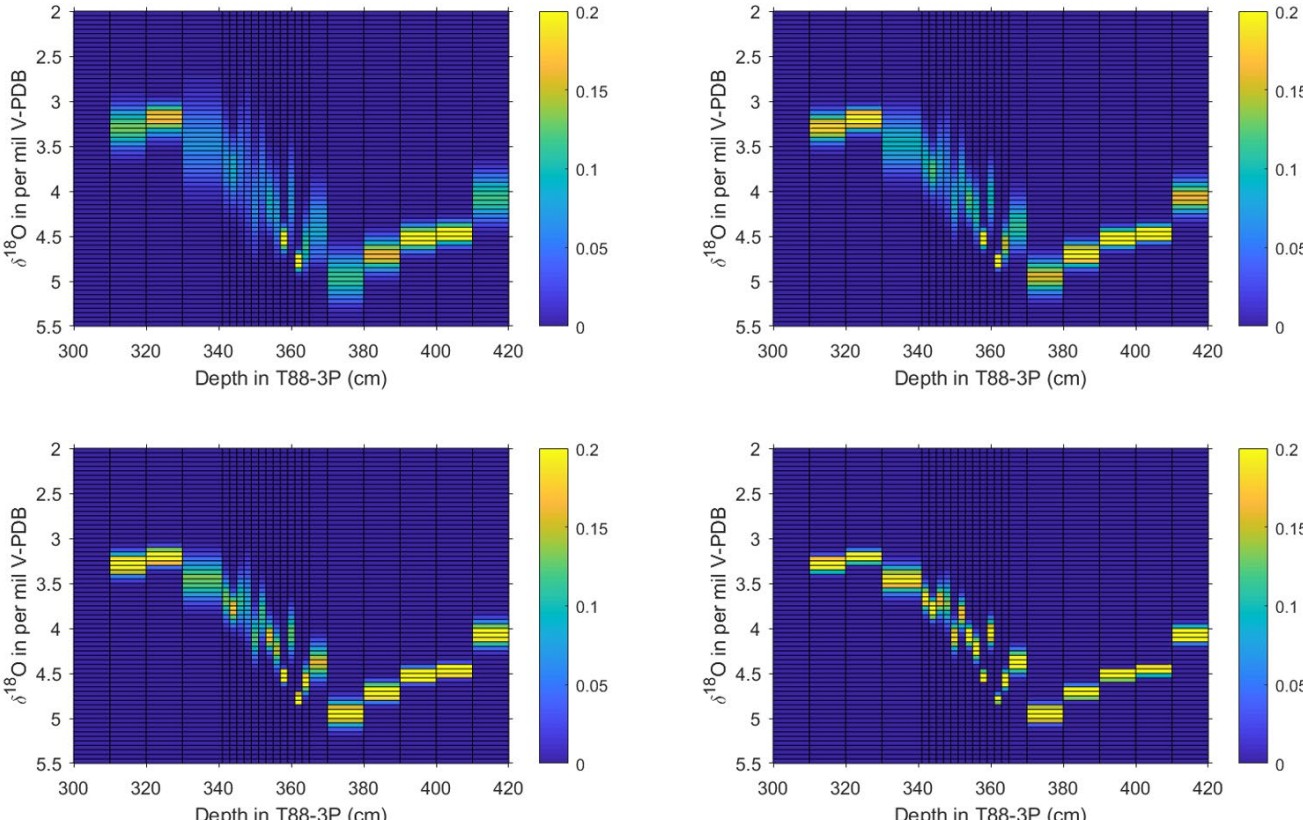

**Figure 9. Output of estimate of pooled specimen variance for T88-3P. Using the unmixed populations of *N. pachyderma* from the deglacial interval, based the upon single shell measurements presented here (Figure 7), a pooled synthetic measurement was created using the probability of each population, and a synthesised normal distribution with the same mean and standard deviation. Estimates were made for (Top left) 5, (Top Right) 10, (Bottom left) 20, and (Bottom Right) 50 specimens per pooled analysis. For each sample 10,000 replicates were produced, plotted here is a 'heat map', the colour represent the probability (counts normalised to 1) of a particular value occurring as a pooled value.**

| Lab Code | | Sample ID | Depth in core (cm) | Species | $^{13}C/^{12}C$ ratio ($\delta^{13}C$ in ‰) | Conventional Radiocarbon Age (in $^{14}C$ yr BP) | ± | Cal Age (in cal. Yr BP) | Cal Age (in cal. Yr BP) |
|---|---|---|---|---|---|---|---|---|---|
| Beta | 343133 | T883P001BULL | 1 | *G. bulloides* | -0.41 | 970 | 30 | 626 | 508 |
| Beta | 343134 | T883P150BULL | 150 | *G. bulloides* | -0.5 | 4230 | 30 | 4418 | 4225 |
| Beta | 343135 | T883P295BULL | 295 | *G. bulloides* | -1.07 | 8570 | 40 | 9363 | 9077 |
| Beta | 343136 | T883P340BULL | 340 | *G. bulloides* | -0.76 | 10990 | 40 | 12651 | 12445 |
| Beta | 343137 | T883P380PACH | 380 | *N. pachyderma* | -0.68 | 21150 | 90 | 25320 | 24600 |
| Beta | 343138 | T883P500BULL | 500 | *G. bulloides* | -0.68 | 37470 | 370 | 41890 | 41360 |

5   **Table 1: Raw and calibrated radiocarbon ages. Conventional radiocarbon age represents the Measured radiocarbon age (see supplementary table 1) corrected for isotopic fraction.**

**Supplementary Information**

| Lab Code | | Sample ID | Depth in core (cm) | Species | Measured Radiocarbon Age | ± | $^{13}C/^{12}C$ ratio (δ$^{13}C$ in ‰) | Conventional Radiocarbon Age (in $^{14}C$ yr BP) | ± | Cal Age (in cal. Yr BP) | Cal Age (in cal. Yr BP) |
|---|---|---|---|---|---|---|---|---|---|---|---|
| Beta | 343133 | T883P001BULL | 1 | *G. bulloides* | 570 | 30 | -0.41 | 970 | 30 | 626 | 508 |
| Beta | 343134 | T883P150BULL | 150 | *G. bulloides* | 3830 | 30 | -0.5 | 4230 | 30 | 4418 | 4225 |
| Beta | 343135 | T883P295BULL | 295 | *G. bulloides* | 8180 | 40 | -1.07 | 8570 | 40 | 9363 | 9077 |
| Beta | 343136 | T883P340BULL | 340 | *G. bulloides* | 10590 | 40 | -0.76 | 10990 | 40 | 12651 | 12445 |
| Beta | 343137 | T883P380PACH | 380 | *N. pachyderma* | 20750 | 90 | -0.68 | 21150 | 90 | 25316 | 24596 |
| Beta | 343138 | T883P500BULL | 500 | *G. bulloides* | 37080 | 370 | -0.68 | 37470 | 370 | 41890 | 41358 |

5    **Supplementary Table 1: Raw and calibrated radiocarbon ages. Conventional radiocarbon age represents the Measured radiocarbon age corrected for isotopic fraction. Calendar ages were determined with the Marine 13 Calibration curve, determined ages in Table 1 have been rounded to the nearest 10 for samples with a standard deviation > 50 (i.e., sample T883P380PACH and T883P500BULL), here they are left unrounded.**

| Depth in core T88-3P (in cm) | GICC05 Aligned Timescale (yr BP)* |
|---|---|
| 0 | 0 |
| 285 | 9332.6 |
| 365 | 12114.8 |
| 405 | 25777 |
| 425 | 29442.4 |
| 475 | 35726.6 |
| 510 | 38190.5 |
| 605 | 44300 |
| 665 | 46804.6 |
| 695 | 48607.5 |
| 745 | 54110.3 |

**Supplementary Table 2: Tie-points used for oxygen isotope tuned age model, based upon tuning to the NGRIP GICC05 timescale (*BP = GICC05 b2k - 50 year).**

30

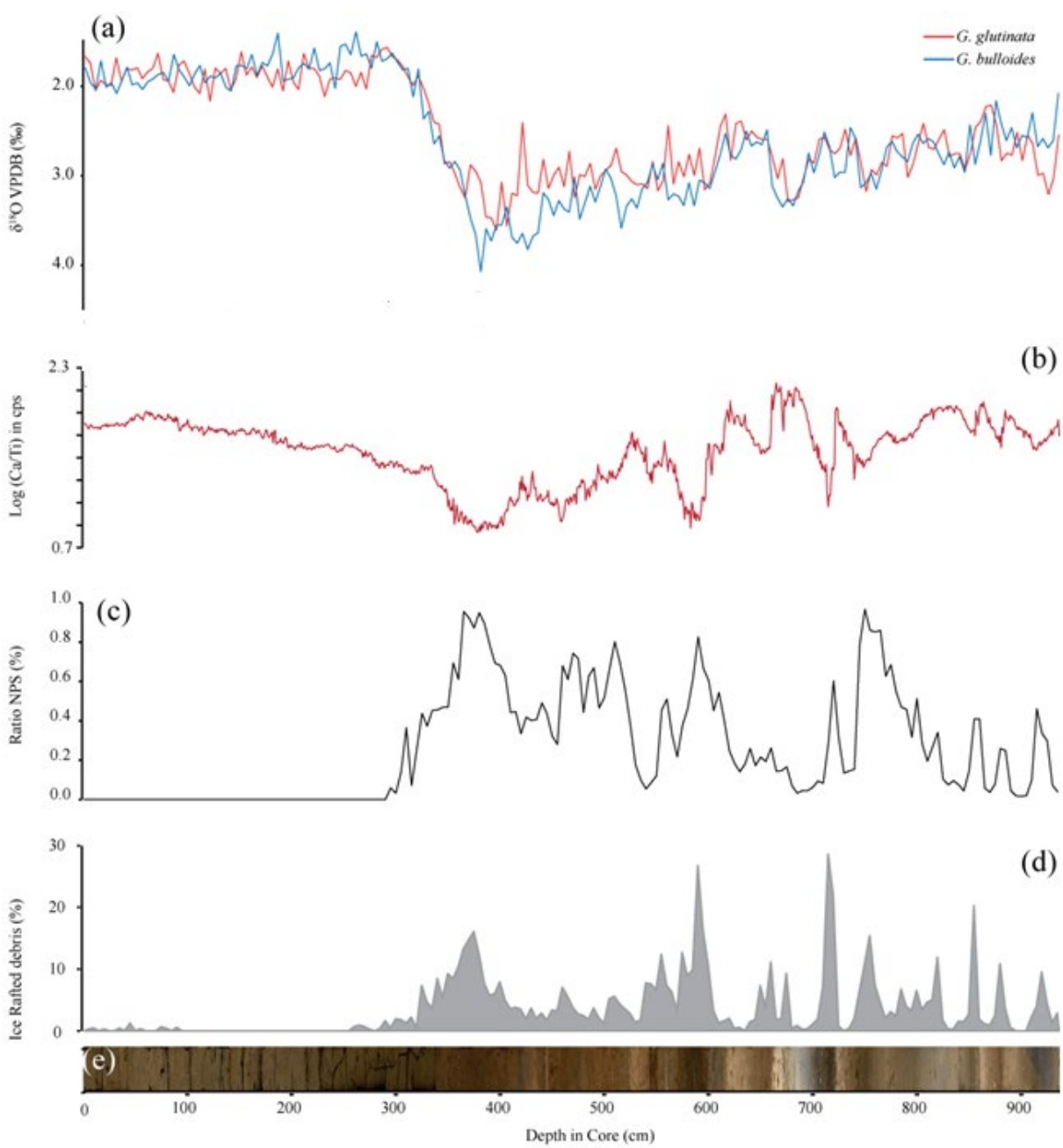

**Supplementary Figure 1: Core stratigraphy of T88-3P with (a) $\delta^{18}O$ of *G. glutinata* (red) and *G. bulloides* (blue), (b) Log(Ca/Ti) ratio (c) abundance ratio (green) of *N. pachyderma* and *G. bulloides* (see methods), (d) percentage of ice rafted debris from particle counts (grey), and (e) Image of core T88-3P. Note the absence of *N. pachyderma* and IRD in the upper 300 cm.**

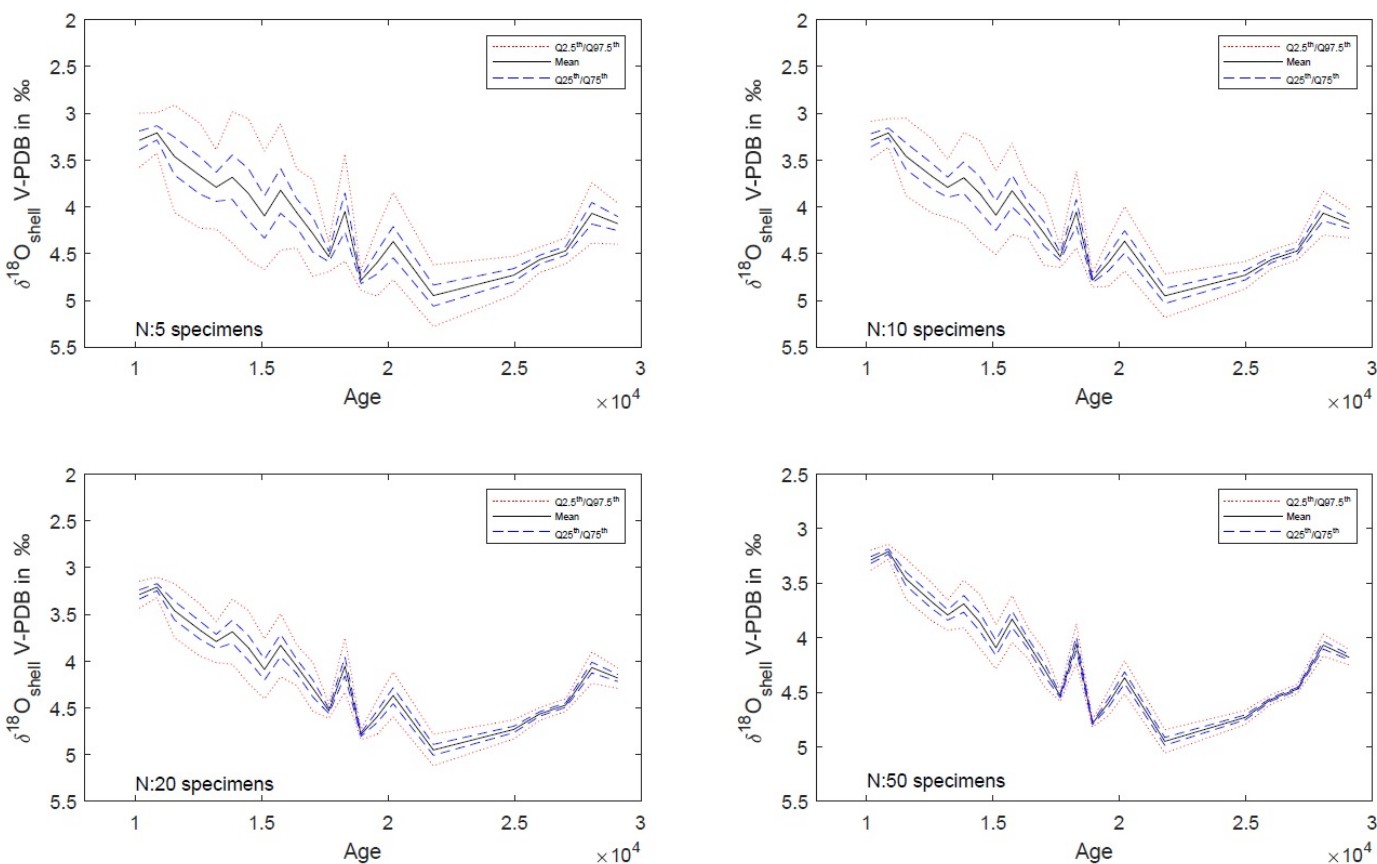

**Supplementary Figure 2: Output of estimate of pooled specimen variance for T88-3P. Using the unmixed populations of *N. pachyderma*, based the upon single shell measurements presented here, a pooled synthetic measurement was created using the probability of each population, and a synthesised normal distribution with the same mean and standard deviation. Estimates were made for 5, 10, 20, and 50 specimens. For each sample 10,000 replicates were produced, the mean (black line) of these pooled specimens remains near constant as a by-product of the number of replicates and therefore the purpose of comparison the quantiles are plotted for each sample against age. Four quantiles are used, the 2.5th and 97.5th quantiles (red dotted line) and the 25th and 75th (blue dashed line), which highlight the spread in the synthesised pooled specimen data.**