# Peer review of "Modal shift in North Atlantic seasonality during the last deglaciation"

_Climate of the Past, 2018_

## Referee Comment (RC1) · Anonymous Referee #1 · 6 May 2019

Brummer and co-authors present single specimen stable isotope measurements of polar species N. pachyderma and transitional species G. bulloides for core T88-3P in the northern mid-latitude North Atlantic. The authors deduce that two different populations of N. pachyderma existed throughout the last deglaciation and that, based on modern observations in the northern North Atlantic, these populations represent calcification during different periods of the year and thus under different environmental conditions. The study provides important new insights and merits publication in a journal like Climate of the Past. However, before the current manuscript could be accepted for publication, there are several points that need to be addressed/explained better and the inconsistencies in labeling etc. need to be corrected. So overall, I am recommending major revisions.

[Figure]

There are three major concerns that I have and which I will outline first.

1) Unimodal mode of G. bulloides and G. bulloides $\delta$13C values

The authors state that the single specimen isotope data of G. bulloides are unimodal, but give not reasoning for this statement. Subsequently, they use the unimodal distribution of G. bulloides as evidence that the two populations of N. pachyderma cannot be related to bioturbation (more on this in point 2). I would like to see some justification for declaring the G. bulloides data unimodal in the text. Whereas the $\delta$18O values show much less scatter than the N. pachyderma data, the respective $\delta$13C data show a range of 0.5‰ at some levels and I wonder, if this is not a reflection of more than one population. This statement is, however, only valid if the $\delta$13C values plotted in Figure 3 are actually correct, because G. bulloides $\delta$13C values should (mostly) be negative and the scale on the Figure is positive and has exactly the same range as for N. pachyderma.

2) Influence of bioturbation

Whereas I agree with the authors in the general sense that the occurrence of two populations cannot be explained by bioturbation, I would urge them to be more careful in those cases where one of the populations is presented by only 1 to 4 specimens. In this regard, it is essential to include an abundance record (which could be the N. pachyderma ratio record from Fig. 2) of both species in Figure 3. Since Figure 2 is presented vs. depth and Figure 3 vs. age, it is impossible for the reader to see where abundance minima of the respective species could have led to a "bias" in the single specimen isotope data (also in G. bulloides during periods of near dominance of N. pachyderma). For example, I do not perceive the argument of the unimodal mode of G. bulloides valid for the two specimens of population 2 in the third line of Table 2 [see note below on correcting column 1 of this table], if that level has already a low abundance of N. pachyderma and can thus be much more likely affected by –even if assumed minor, i.e. over 5 instead of 10 or 20 cm depth– bioturbation. In addition,

Figure 3 should include a plot showing the variations in the sediment rates, so that the reader can see where low sedimentation rates might have increased the chance of bioturbational mixing. Including these plots might not change the story, but provides the reader with the option to judge him/herself in which levels bioturbation might have affected the single specimen data (and to what degree) or not.

3) Age model and 14C calibration

The authors made the effort to test different approaches to establish an age model, but in the end the reader does not know, which age model/age control points were used to produce the record of the data vs. age as shown in Figure 3. So please, specify this and provide either in the main manuscript or in the supplementary material a table listing the final age control points. Did you combine? If yes, did you then discard some calibrated ages? Issues with the text and information in Table 1 regarding the 14C calibration: Table 1 and section 2.4 and supplementary material: your measured age should be the same as the conventional age, i.e. the raw 14C concentration converted into an uncorrected 14C age (using the Libby half-life). If you calibrate with Marine13 this uncorrected age would be the one used to calibrate. So I do not understand how your Table 1 can list conventional ages that are 400 years higher than the measured age –which to me looks like a reservoir age correction going into the wrong direction! And I am not sure, which age –measured or conventional– was actually calibrated! If you analyze marine material like foraminifera the measured/conventional age needs to be corrected for the reservoir effect, i.e. transferred to "atmospheric 14C levels" by subtracting the reservoir age (such as 400 yr), if you want to calibrate with atmospheric level calibration data like Intcal13. Since you are calibrating with Marine13 you do not use a fixed reservoir age (of 400 years)! During the Holocene (0-10.5 cal ka BP) section the reservoir age is provided as outcome of the ocean-atmosphere box diffusion model and varies "significantly" over time –see for example Figure 4b in Hughen et al. 2004 on Marine04. In the glacial section, where a fixed reservoir age is used, the value is 405 years and not 400 years (see p. 1877 in Reimer et al. 2013). Inconsistency

between p. 3 line 30, supplementary material: you state that the deepest/oldest 14C age was not used/excluded; so why it is then shown and used in Figure S2?

While correcting the 14C calibration will change the age model, this will not affect the general conclusions of the manuscript.

Additional comments:

Main manuscript p. 3 abundance counts: please specify a) how the % IRD was calculated; b) why a Ratio of NPS was calculated and not the more commonly used % N. pachyderma.

p. 3 Stable isotope section: please mention a) the resolution at which the single specimen measurements were done (4 cm?); b) if the N. pachyderma specimens were encrusted; c) which are the international carbonate standards used during the stable isotope analyses?

p. 3 core stratigraphy (besides comments above on 14C calibration): may be specify that you follow Reimer et al. (2013) when using $\Delta$R of 0$\pm$200 yr. line 29-30: if you keep the sentence, specify which sample was excluded (do not assume that every reader will read the supplementary material in detail). line 31-32: how many specimens of G. bulloides and G. glutinata were analyzed for the "bulk" analyses? line 35: include that the tuning was done to the $\delta$18O record of NGRIP, which, I assume, is presented on the GICC05 chronology. If you used NGRIP on GICC05, did you remember to correct the GICC05 b2k ages to BP ages (by subtracting 50 years) to make the tuned ages compatible with the calibrated 14C ages? line 36-37: you are providing information on temporal resolution and not sedimentation rates. I do not find this very informative and would like to see a figure showing the variations. Also, the sentence in its current phrasing is incomplete.

p. 4 line 4: what does IFA stand for?

p. 4 line 20: year missing for Jonkers and Kucera reference

p. 5 line 14-15: what about within glacial mixing/bioturbation?

p. 6 line 35: N. pachyderma $\delta$18O data not shown in Figure 2.

Table 1: following the recommendations of Stuiver & Reimer " Users are advised to round results to the nearest 10 yr for samples with standard deviation in the radiocarbon age greater than 50 yr".

Table 2: first column: please correct; what you are listing are not or incomplete depths. since the data itself is not shown vs. depth, it would be good to have an age column as well. Reduce the number of decimal places in the Prob and Mean columns, so that the numbers become easier to read.

Figure 3, 4, S1 etc.: in all the axis label referring to the NGRIP $\delta$18O data, replace the "SW (sea water ??)" by "ice". Provide reference for NGRIP data in figure captions.

Figure 3: as mentioned already above under point 1, correct the $\delta$13C scale for G. bulloides.

Figure S4: the right panel does not show the filtered NGRIP record = tuning target. Why is the SPECMAP error applied and not the GICC05 errors?

Supplementary material text: line 24 insert $\delta$18O before ice core and mention that the NGRIP record is on the GICC05 time scale.

line 27: provide more information on the "simple filter". for which frequencies did you filter and why?

---

## Referee Comment (RC2) · Anonymous Referee #2 · 10 Jun 2019

Review of the manuscript "Modal shift in North Atlantic seasonality during the last deglaciation" by Brummer et al. The authors present a study using single specimen isotopes on the planktonic foraminifera G. bulloides and N. pachyderma to show that during the deglaciation in the North Atlantic two different populations of pachyderma, one in spring and one in late-summer, occurred, while only one existed during the glacial and the Holocene. This variation would not have been possible to resolve using traditional pooled specimen analyses. These results suggest that these two populations are reflective of modern conditions from the present Irminger Sea further to the north, where sediment trap data for pachyderma show a similar double abundance peak. An interesting implication of the results is that when the pooled specimen record is reflecting a change of population rather than presenting the same signal, does this

imply that no deglacing warming took place in this area of the North Atlantic? This study is a very interesting application of single foraminifer analyses of stable isotopes showing the use of single foraminifer analyses, highlighting the increasing attention it receives in the literature. The manuscript is mostly clearly written and easy to follow.

My main issue with the study is that the number of analyses, i.e. specimens, per sample is too low to give a representative split up in different populations. Up to 20 specimens were picked per sample, and for quite a few samples less than that were successfully analysed. What is the risk that the split into two populations for these samples is not simply due to highly variable values that only give the impression of separate populations?

Page 1 Line 24: are you suggesting the deglaciation lasted for 10 kyr? Line 32: many more references could be cited here to better reflective the literature. These references are all from the same lab.

Page 2, Line 29: delete the first "and"

Page 3, 2.3 title: add single specimens to it to distinguish from 2.4 where the bulk analyses are described. Line 21: the pachydermas weighed >10 $\mu$g? 2.4, line 24: how many specimens/what weight were used? Line 37: "varoes"

Page 4, line 20: missing year in Jonkers and Kucera Line 32: I assume these are the pooled d18O? Line 36: "during IRD events"

Page 5, line 4: The striking bimodality is quite difficult to see, it could simply be more variation in the analyses. Why not plot the results also as histograms? And similar for the d13C results; it is not easy to see now how the variations are. Additionally, why is the x-axis labelled in x time 10 4 years? This is confusing, just stick to the regular ka.

Page 6, line 7: Is 250-300 $\mu$m correct? Line 8: were any of the sediment-trap pachydermas genetically determined? Line 35: pachyderma is also unlikely to have lived in this meltwater; they normally stick below this relatively fresh layer.

Page 7, line 31: delete "."

Page 8, line 7: the Bard, 2001 reference is missing from the References Section 4.3: the results here show that in a setting like the North Atlantic the pooled specimen analyses may be biased when not enough specimens are being used. Could you provide an estimate how many specimens would be needed to give a reliable estimate?

Figure 2b: Is this 14C age of 41900 years used for the age model or not? It seems not, so then it should be deleted from the figure or indicated as such. Figure 5: Add headings of the different areas on top of each "column".

To sum up, this manuscript is very suitable for Climate of the Past using a technique that is receiving more and more application. The manuscript illustrates the opportunity of single foraminifer analyses. After the authors have especially dealt with the number of specimens used per analyses and the minor comments, I see no further issues with this manuscript being published.

---

## Author Response (AR1)

Dear Prof. Marit-Solveig Seidenkrantz,

Outlined below in brief are the changes to document:

- Table 2 has been removed and will, following publication or prior to be uploaded as a supplement. This is because we now have two intervals of two species (= 4 additional tables) worth of data;
- We have added in SAR and abundance
- We have added in raw d18O plots, inc. box-plots that have the raw and outlier corrected data
- We have corrected the error in radiocarbon dates, and have attached to the peer-review file a copy of Beta analytics analysis
- We have removed the time plots – whilst we have included an explanation of both age models (a radiocarbon and stable isotope stratigraphy that is independent of each other)
- We have removed the carbon isotope data
- We have changed the probability data to an easier figure

[Figure]

**Beta Analytic Inc.**
4985 SW 74 Court
Miami, Florida 33155 USA
Tel: 305 667 5167
Fax: 305 663 0964
Beta@radiocarbon.com
www.radiocarbon.com

**Darden Hood**
President

**Ronald Hatfield**
**Christopher Patrick**
Deputy Directors

*Consistent Accuracy . . .*
*. . . Delivered On-time*

February 28, 2013

Dr. Wouter Feldmeijer
VU University
Earth and Climate Cluster
de Boelelaan 1085
HV Amsterdam, 1081
The Netherlands

RE: Radiocarbon Dating Results For Samples T883P001BULL, T883P150BULL, T883P295BULL, T883P340BULL, T883P380PACH, T883P500BULL

Dear Dr. Feldmeijer:

Enclosed are the radiocarbon dating results for six samples recently sent to us. They each provided plenty of carbon for accurate measurements and all the analyses proceeded normally. The report sheet contains the dating result, method used, material type, applied pretreatment and two-sigma calendar calibration result (where applicable) for each sample.

This report has been both mailed and sent electronically, along with a separate publication quality calendar calibration page. This is useful for incorporating directly into your reports. It is also digitally available in Windows metafile (.wmf) format upon request. Calibrations are calculated using the newest (2004) calibration database. References are quoted on the bottom of each calibration page. Multiple probability ranges may appear in some cases, due to short-term variations in the atmospheric 14C contents at certain time periods. Examining the calibration graphs will help you understand this phenomenon. Calibrations may not be included with all analyses. The upper limit is about 20,000 years, the lower limit is about 250 years and some material types are not suitable for calibration (e.g. water).

We analyzed these samples on a sole priority basis. No students or intern researchers who would necessarily be distracted with other obligations and priorities were used in the analyses. We analyzed them with the combined attention of our entire professional staff.

Information pages are enclosed with the mailed copy of this report. They should answer most of questions you may have. If they do not, or if you have specific questions about the analyses, please do not hesitate to contact us. Someone is always available to answer your questions.

The cost of analysis was previously invoiced. As always, if you have any questions or would like to discuss the results, don't hesitate to contact me.

Sincerely,

*Darden Hood*

**Digital signature on file**

[Figure]

**REPORT OF RADIOCARBON DATING ANALYSES**

Dr. Wouter Feldmeijer

VU University

Report Date: 2/28/2013

Material Received: 2/19/2013

| Sample Data | Measured Radiocarbon Age | 13C/12C Ratio | Conventional Radiocarbon Age(*) |
|---|---|---|---|
| Beta - 343133
SAMPLE : T883P001BULL
ANALYSIS : AMS-Standard delivery
MATERIAL/PRETREATMENT : (foraminifera): none
2 SIGMA CALIBRATION : Cal AD 1330 to 1440 (Cal BP 620 to 510) | 570 +/- 30 BP | -0.41 o/oo | 970 +/- 30 BP |
| Beta - 343134
SAMPLE : T883P150BULL
ANALYSIS : AMS-Standard delivery
MATERIAL/PRETREATMENT : (foraminifera): none
2 SIGMA CALIBRATION : Cal BC 2450 to 2280 (Cal BP 4400 to 4240) | 3830 +/- 30 BP | -0.5 o/oo | 4230 +/- 30 BP |
| Beta - 343135
SAMPLE : T883P295BULL
ANALYSIS : AMS-Standard delivery
MATERIAL/PRETREATMENT : (foraminifera): none
2 SIGMA CALIBRATION : Cal BC 7370 to 7150 (Cal BP 9320 to 9100) | 8180 +/- 40 BP | -1.07 o/oo | 8570 +/- 40 BP |
| Beta - 343136
SAMPLE : T883P340BULL
ANALYSIS : AMS-Standard delivery
MATERIAL/PRETREATMENT : (foraminifera): none
2 SIGMA CALIBRATION : Cal BC 10640 to 10570 (Cal BP 12590 to 12520) AND Cal BC 10500 to 10440 (Cal BP 12450 to 12390) | 10590 +/- 40 BP | -0.76 o/oo | 10990 +/- 40 BP |

Dates are reported as RCYBP (radiocarbon years before present, "present" = AD 1950). By international convention, the modern reference standard was 95% the 14C activity of the National Institute of Standards and Technology (NIST) Oxalic Acid (SRM 4990C) and calculated using the Libby 14C half-life (5568 years). Quoted errors represent 1 relative standard deviation statistics (68% probability) counting errors based on the combined measurements of the sample, background, and modern reference standards. Measured 13C/12C ratios (delta 13C) were calculated relative to the PDB-1 standard.

The Conventional Radiocarbon Age represents the Measured Radiocarbon Age corrected for isotopic fractionation, calculated using the delta 13C. On rare occasion where the Conventional Radiocarbon Age was calculated using an assumed delta 13C, the ratio and the Conventional Radiocarbon Age will be followed by "*". The Conventional Radiocarbon Age is not calendar calibrated. When available, the Calendar Calibrated result is calculated from the Conventional Radiocarbon Age and is listed as the "Two Sigma Calibrated Result" for each sample.

[Figure]

**BETA ANALYTIC INC.**

DR. M.A. TAMERS and MR. D.G. HOOD

4985 S.W. 74 COURT
MIAMI, FLORIDA, USA 33155
PH: 305-667-5167 FAX:305-663-0964
beta@radiocarbon.com

**REPORT OF RADIOCARBON DATING ANALYSES**

Dr. Wouter Feldmeijer

Report Date: 2/28/2013

| Sample Data | Measured Radiocarbon Age | 13C/12C Ratio | Conventional Radiocarbon Age(*) |
|---|---|---|---|
| | | | |

Beta - 343137           20750 +/- 90 BP      -0.68 o/oo      21150 +/- 90 BP
SAMPLE : T883P380PACH
ANALYSIS : AMS-Standard delivery
MATERIAL/PRETREATMENT : (foraminifera): none
2 SIGMA CALIBRATION :     Cal BC 23030 to 22530 (Cal BP 24980 to 24480)
* * *
Beta - 343138           37080 +/- 370 BP      -0.98 o/oo      37470 +/- 370 BP
SAMPLE : T883P500BULL
ANALYSIS : AMS-Standard delivery
MATERIAL/PRETREATMENT : (foraminifera): none
2 SIGMA CALIBRATION :     Cal BC 40420 to 39480 (Cal BP 42370 to 41430)
* * *
Dates are reported as RCYBP (radiocarbon years before present, "present" = AD 1950). By international convention, the modern reference standard was 95% the 14C activity of the National Institute of Standards and Technology (NIST) Oxalic Acid (SRM 4990C) and calculated using the Libby 14C half-life (5568 years). Quoted errors represent 1 relative standard deviation statistics (68% probability) counting errors based on the combined measurements of the sample, background, and modern reference standards. Measured 13C/12C ratios (delta 13C) were calculated relative to the PDB-1 standard.

The Conventional Radiocarbon Age represents the Measured Radiocarbon Age corrected for isotopic fractionation, calculated using the delta 13C. On rare occasion where the Conventional Radiocarbon Age was calculated using an assumed delta 13C, the ratio and the Conventional Radiocarbon Age will be followed by "*". The Conventional Radiocarbon Age is not calendar calibrated. When available, the Calendar Calibrated result is calculated from the Conventional Radiocarbon Age and is listed as the "Two Sigma Calibrated Result" for each sample.

**CALIBRATION OF RADIOCARBON  AGE TO CALENDAR YEARS**

(Variables: C13/C12=-0.41:Delta-R=0±0:Glob res=-200 to 500:lab. mult=1)

**Laboratory number:**  **Beta-343133**

**Conventional radiocarbon age:**  **970±30 BP**

**(local reservoir correction not applied)**

**2 Sigma calibrated result:**  **Cal AD 1330 to 1440 (Cal BP 620 to 510)**
**(95% probability)**

Intercept data

Intercept of radiocarbon age
with calibration curve:  Cal AD 1410 (Cal BP 540)

1 Sigma calibrated result:  Cal AD 1360 to 1430 (Cal BP 590 to 520)
(68% probability)

[Figure]

970±30 BP (970±30 adjusted)

Foraminifera

References:

*Database used*
MARINE09
*References to INTCAL09 database*
*Heaton,et.al.,2009, Radiocarbon 51(4):1151-1164, Reimer,et.al, 2009, Radiocarbon 51(4):1111-1150,*
*Stuiver,et.al,1993, Radiocarbon 35(1):137-189, Oeschger,et.al.,1975,Tellus 27:168-192*
*Mathematics used for calibration scenario*
*A Simplified Approach to Calibrating C14 Dates*
*Talma, A. S., Vogel, J. C., 1993, Radiocarbon 35(2):317-322*

**Beta Analytic Radiocarbon Dating Laboratory**

*4985 S.W. 74th Court, Miami, Florida 33155 • Tel: (305)667-5167 • Fax: (305)663-0964 • E-Mail: beta@radiocarbon.com*

**CALIBRATION OF RADIOCARBON  AGE TO CALENDAR YEARS**

(Variables:  C13/C12=-0.5:Delta-R=0±0:Glob res=-200 to 500:lab. mult=1)

**Laboratory number:**  **Beta-343134**

**Conventional radiocarbon age:**  **4230±30 BP**

**(local reservoir correction not applied)**

**2 Sigma calibrated result:**  **Cal BC 2450 to 2280 (Cal BP 4400 to 4240)**
**(95% probability)**

Intercept data

Intercept of radiocarbon age
with calibration curve:  Cal BC 2400 (Cal BP 4340)

1 Sigma calibrated result:  Cal BC 2440 to 2330 (Cal BP 4390 to 4280)
(68% probability)

[Figure]

4230±30 BP (4230±30 adjusted)                                                Foraminifera

**Mathematics used for calibration scenario**
    A Simplified Approach to Calibrating C14 Dates
    Talma, A. S., Vogel, J. C., 1993, Radiocarbon 35(2):317-322

**Beta Analytic Radiocarbon Dating Laboratory**

*4985 S.W. 74th Court, Miami, Florida 33155 • Tel: (305)667-5167 • Fax: (305)663-0964 • E-Mail: beta@radiocarbon.com*

**CALIBRATION OF RADIOCARBON  AGE TO CALENDAR YEARS**

(Variables:  C13/C12=-1.07:Delta-R=0±0:Glob res=-200 to 500:lab. mult=1)

**Laboratory number:**   **Beta-343135**

**Conventional radiocarbon age:**   **8570±40 BP**

**(local reservoir correction not applied)**

**2 Sigma calibrated result:**   **Cal BC 7370 to 7150 (Cal BP 9320 to 9100)**
**(95% probability)**

Intercept data

Intercept of radiocarbon age
with calibration curve:   Cal BC 7280 (Cal BP 9240)

1 Sigma calibrated result:   Cal BC 7320 to 7220 (Cal BP 9270 to 9170)
(68% probability)

[Figure]

**CALIBRATION OF RADIOCARBON AGE TO CALENDAR YEARS**

(Variables: C13/C12=-0.76:Delta-R=0±0:Glob res=-200 to 500:lab. mult=1)

**Laboratory number:** **Beta-343136**

**Conventional radiocarbon age:** **10990±40 BP**

**(local reservoir correction not applied)**

**2 Sigma calibrated results:** **Cal BC 10640 to 10570 (Cal BP 12590 to 12520) and**
**(95% probability)** **Cal BC 10500 to 10440 (Cal BP 12450 to 12390)**

Intercept data

Intercept of radiocarbon age
with calibration curve: Cal BC 10610 (Cal BP 12560)

1 Sigma calibrated results: Cal BC 10620 to 10590 (Cal BP 12580 to 12540) and
(68% probability) Cal BC 10480 to 10470 (Cal BP 12430 to 12420)

[Figure]

**CALIBRATION OF RADIOCARBON AGE TO CALENDAR YEARS**

(Variables: C13/C12=-0.68:Delta-R=0±0:Glob res=-200 to 500:lab. mult=1)

**Laboratory number:** **Beta-343137**

**Conventional radiocarbon age:** **21150±90 BP**

**(local reservoir correction not applied)**

**2 Sigma calibrated result:** **Cal BC 23030 to 22530 (Cal BP 24980 to 24480)**
**(95% probability)**

Intercept data

Intercept of radiocarbon age
with calibration curve: Cal BC 22840 (Cal BP 24800)

1 Sigma calibrated result: Cal BC 22970 to 22580 (Cal BP 24920 to 24530)
(68% probability)

[Figure]

**CALIBRATION OF RADIOCARBON  AGE TO CALENDAR YEARS**

(Variables:  C13/C12=-0.98:Delta-R=0±0:Glob res=-200 to 500:lab. mult=1)

**Laboratory number:**  **Beta-343138**

**Conventional radiocarbon age:**  **37470±370 BP**

**(local reservoir correction not applied)**

**2 Sigma calibrated result:**  **Cal BC 40420 to 39480 (Cal BP 42370 to 41430)**
**(95% probability)**

Intercept data

Intercept of radiocarbon age
with calibration curve:  Cal BC 39960 (Cal BP 41910)

1 Sigma calibrated result:  Cal BC 40180 to 39740 (Cal BP 42130 to 41690)
(68% probability)

[revised manuscript text omitted]

**EDITOR COMMENT:**

Comments to the Author:
Dear Drs. Brummer and Metcalfe,

Thank you for re-submitting your manuscript "Modal shift in North Atlantic seasonality during the last deglaciation" to "Climate of the Past". As you are aware, your manuscript has been evaluated by two reviewers, who are overall positive, although they also raise some issues to be dealt with. I am pleased to see your detailed answers to the reviewers' comments, and your plans for corrections of your manuscript.

I thus invite you to submit a revised version of your manuscript taking all review comments into accounts. Please note that if you choose to resubmit, your revised version of the manuscript may be sent for a second round of reviews.

As always, if you choose to resubmit, please reply to all reviewers' comments in detail and mark clearly any changes made to the manuscript in the text through highlights or track-changes.

Kind regards,
Marit-Solveig Seidenkrantz
co-Editor in Chief, Climate of the Past

**Response to Referee 1 – Brummer et al., "*Modal shift in North Atlantic seasonality during the last deglaciation*"**

We thank the reviewer for raising some important points. As the reviewer notes our conclusions are not altered by our age model however upon reflection we agree with the reviewer and therefore would like to take the opportunity to expand our manuscript's age-depth model (including adding SAR) in a revised Manuscript (we thank the reviewer for drawing attention to the oddity in table 1 and will correct this). We also thank the reviewer for noting that our $\delta^{13}$C of *G. bulloides* is actually normalised between 0 and 1. The 'normalisation' has kept the difference between the absolute values, the data is just presented on a relative scale. We will correct this is in a revised MS (although we present the absolute values in a plot below). In the following, reviewer comments are in RED, our responses are in BLACK:

There are three major concerns that I have and which I will outline first.

1) Unimodal mode of G. bulloides and G. bulloides $\delta$13C values

The authors state that the single specimen isotope data of G. bulloides are unimodal, but give not reasoning for this statement. Subsequently, they use the unimodal distribution of G. bulloides as evidence that the two populations of N. pachyderma cannot be related to bioturbation (more on this in point 2). I would like to see some justification for declaring the G. bulloides data unimodal in the text. Whereas the $\delta$18O values show much less scatter than the N. pachyderma data, the respective $\delta$13C data show a range of 0.5‰ at some levels and I wonder, if this is not a reflection of more than one population.

We did not produce a similar figure of the unimodal nature of *G. bulloides*, however at the reviewer's suggestion we have performed this and there are at some depths indeed more than one population – this is an equally interesting result and we will add this to a revised manuscript. We thank the reviewer for their suggestion – our focus was on *N. pachyderma* for this analysis because as a polar species it should have a reduced ecological range and hence our interest in more than one population. A quick figure of the *G. bulloides* data is plotted here:

[Figure]

In addition, we will add in values of glacial *N. pachyderma* and *G. bulloides* from much deeper in the core which we can add as a comparison between ecological change across the deglaciation and a glacial interval.

It is important to clarify that: "Subsequently, they use the unimodal distribution of G. bulloides as evidence that the two populations of N. pachyderma cannot be related to bioturbation" our exclusion of bioturbation is not only based upon our perception of the unimodality of *G. bulloides* but as we state further down in the same section: "*However, we exclude this particular scenario because sedimentary features (Fig. 2) indicate a lack of discernible mixing, i.e. the sharpness of the IRD percentage, the Log(Ca/Ti) and the percentage of NPS all indicate that bioturbation is at a minimum*". And hence (This statement is, however, only valid) is not the sole reason for whether or not our statement is valid, although it is a strong argument.

This statement is, however, only valid if the $\delta$13C values plotted in Figure 3 are actually correct, because G. bulloides $\delta$13C values should (mostly) be negative and the scale on the Figure is positive and has exactly the same range as for N. pachyderma.

Apologies, we thank the reviewer for pointing out this mistake. Whilst, the values were correct, the plotting tool had rescaled the colour scale to values between 0 and 1. We will correct this in a revised version, but for now we present the data not scaled.

[Figure]

**Age vs. $\delta^{18}O_{shell}$ (filled colours $\delta^{13}C_{shell}$)**

2) Influence of bioturbation

Whereas I agree with the authors in the general sense that the occurrence of two populations cannot be explained by bioturbation, I would urge them to be more careful in those cases where one of the populations is presented by only 1 to 4 specimens.

We agree, hence why we sought to give the reader alternative explanations as well, i.e., section 4.2 onwards. There are only 3 populations with less than < 3 specimens; only 5 populations with less than < 4 specimens. We will add in the following text: "However, it is important to note that for several depths in core this second population may only represent a few specimens ($n_{< 3\ specimens}$ = 3; and $n_{< 4\ specimens}$ = 5)"

In this regard, it is essential to include an abundance record (which could be the N. pachyderma ratio record from Fig. 2) of both species in Figure 3. Since Figure 2 is presented vs. depth and Figure 3 vs. age, it is impossible for the reader to see where abundance minima of the respective species could have led to a "bias" in the single specimen isotope data (also in G. bulloides during periods of near dominance of N. pachyderma).

Although we did present both (top panel) depth and (bottom panel) age in figure 4, we can certainly add additional panels into the figures to highlight the abundance of the species. We will plot the abundance data also on both age and depth scales.

For example, I do not perceive the argument of the unimodal mode of G. bulloides valid for the two specimens of population 2 in the third line of Table 2 [see note below on correcting column 1 of this table], if that level has already a low abundance of N. pachyderma and can thus be much more likely affected by –even if assumed minor, i.e. over 5 instead of 10 or 20 cm depth– bioturbation.

We thank the reviewer for their comment, though this is why we state, "alternative scenarios that give the same or a similar solution for the existence of two populations can be envisaged". Whilst we have explained (section 4.2.3) how bioturbation would potentially affect our observations down core – we can present the abundance data that we do have and include a discussion of the abundance of foraminifera with respect to bioturbation.

In addition, Figure 3 should include a plot showing the variations in the sediment rates, so that the reader can see where low sedimentation rates might have increased the chance of bioturbational mixing. Including these plots might not change the story, but provides the reader with the option to judge him/herself in which levels bioturbation might have affected the single specimen data (and to what degree) or not.

We have a sparse number of tie-points, which is why we did not plot the sedimentation rate in one or more panels (in the background). A sparse number of tie points may give a spurious impression, for instance at times of high or low IRD the SAR may vary considerably yet with a sparse number of tie points we have only 'book-ended' these results with a single SAR value. Figure 4, in which one panel has age and the other depth was our attempt around presenting the depth to the reader (as the reviewer suggests). Here the reader can see the two populations vs depth in core and therefore can for themselves consider the mixed layer or bioturbation depth which may or may not vary between 5-15 cm. With the reviewer's suggestion of expanding figure 4 (see comment above) we hope that those changes will be sufficient.

Additional comments:

Main manuscript p. 3 abundance counts: please specify a) how the % IRD was calculated; b) why a Ratio of NPS was calculated and not the more commonly used % N. pachyderma.

IRD was calculated from a sum total of foraminifera and IRD. This does have complications for the calculation of % N. pachyderma as it is a closed sum with some variation due to changes in IRD. Whilst this is less than ideal, unpublished data comparing these methodologies shows that the % N. pachyderma produced from a sum of foraminifera and IRD is consistent with %N. pachyderma as a sum of only foraminifera, when IRD is less than ~50% of the total grains. Higher values of IRD will, of course, alter this.

p. 3 Stable isotope section: please mention a) the resolution at which the single specimen measurements were done (4 cm?); b) if the N. pachyderma specimens were encrusted; c) which are the international carbonate standards used during the stable isotope analyses?

We performed faunal counts every 4 cm (line 2, pg 3). We will clarify that this spacing is different for the isotopes. Therefore, we intend to alter pg. 3, line 2 as follows:

"The core sections of the entire working half were sampled every cm, resulting in 1 cm sample slices that were each washed over a 63 µm sieve mesh, dried overnight at ~75°C and subsequently size fractionated into 63-150 µm and >150 µm. For abundance counts of planktonic foraminifera, slices every 4 cm were used, the counts were performed on…"

We will alter pg. 3 line 13 to: "Slices for isotope analysis were selected first at 10 cm resolution and then at specific sections down core every 2 cm. For each slice 20 shells of both left coiling *N. pachyderma* and *G. bulloides* were picked at random from the 250 - 300 µm size fraction (Figs. 2-4)."

p. 3 core stratigraphy (besides comments above on 14C calibration): may be specify that you follow Reimer et al. (2013) when using ΔR of 0±200 yr.

We will repeat the reference at the end of the sentence, so that sentence will read: ", using the Marine13 Calibration curve (Reimer et al., 2013) and a reservoir age of 400 14C years with an error of 200 14C years, expressed mathematically as ΔR: 0 ± 200 14C yr (Reimer et al., 2013)."

line 29-30: if you keep the sentence, specify which sample was excluded (do not assume that every reader will read the supplementary material in detail).

We will make a note in the table to show which is excluded.

line 31-32: how many specimens of G. bulloides and G. glutinata were analyzed for the "bulk" analyses?

The data is based upon the mean of 2 groups (comprised of 5-10 specimens per group) for each species – we will reiterate this in the paper.

line 35: include that the tuning was done to the $\delta$18O record of NGRIP, which, I assume, is presented on the GICC05 chronology. If you used NGRIP on GICC05, did you remember to correct the GICC05 b2k ages to BP ages (by subtracting 50 years) to make the tuned ages compatible with the calibrated 14C ages?

We will adjust the figures accordingly, as stated in the supplement we use an earlier chronology – we will therefore replot and alter the age model according to the GICC05 chronology (this shifts the age ever so slightly).

line 36-37: you are providing information on temporal resolution and not sedimentation rates. I do not find this very informative and would like to see a figure showing the variations. Also, the sentence in its current phrasing is incomplete.

We will add a figure and complete the sentence.

p. 4 line 4: what does IFA stand for?

IFA stands for Individual foraminiferal analysis – we shall, alter the header to: "1.1. Seasonality and single foraminiferal analysis (SFA)" and then alter the header of 2.5 to the same acronym.

p. 4 line 20: year missing for Jonkers and Kucera reference

We will add the date

p. 5 line 14-15: what about within glacial mixing/bioturbation?

The detection of bioturbation is intrinsically related to the difference between two samples, if two samples with uniform values between them are mixed, then it would be impossible to distinguish them, though it does not mean it does not exist. We will clarify this

p. 6 line 35: N. pachyderma $\delta$18O data not shown in Figure 2.

Indeed. Consequently we will change "…of either N. pachyderma or G. bulloides (Fig. 2)…" into "…of either G. bulloides (Fig. 2) or N. pachyderma (Fig. 4)…"

Table 1: following the recommendations of Stuiver & Reimer " Users are advised to round results to the nearest 10 yr for samples with standard deviation in the radiocarbon age greater than 50 yr".

We will round these numbers for the main text and add a supplementary table of unrounded numbers (whilst we agree with the referee we also wish to allow for future readers to know the actual number used).

Table 2: first column: please correct; what you are listing are not or incomplete depths. since the data itself is not shown vs. depth, it would be good to have an age column as well. Reduce the number of decimal places in the Prob and Mean columns, so that the numbers become easier to read.

We will alter this accordingly (it was the sample ID). We will also round the mean, standard deviation and prob numbers to 2 decimal places.

Figure 3, 4, S1 etc.: in all the axis label referring to the NGRIP $\delta$18O data, replace the "SW (sea water ??)" by "ice". Provide reference for NGRIP data in figure captions.

We will alter to just $\delta^{18}O$ as VSMOW is already indicative of the substance.

Figure 3: as mentioned already above under point 1, correct the $\delta$13C scale for G. bulloides.

Altered accordingly.

Inconsistency between p. 3 line 30, supplementary material: you state that the deepest/oldest 14C age was not used/excluded; so why it is then shown and used in Figure S2

We will clarify in a revised MS; we exclude it not for it being incorrect or wrong but due to the fact that the calibration curve at the older end is based upon 'noisy' data (this is not critique, rather it is 'the best of a bad lot' and just a comment on the underlying data used in the construction of the calibration curve) therefore whilst its exclusion as a tie-point in an age model is circumspect it can still be used as an indicator that the age model appears to be 'working' (it is not blank, for instance, therefore the pooled age is likely not older than 50,000 years).

▪▪▪▪▪▪▪▪▪▪▪▪▪▪▪▪▪▪▪▪▪▪▪▪▪▪▪▪▪▪▪▪▪▪▪▪▪▪▪▪▪▪▪▪▪▪▪▪▪▪▪▪▪▪▪▪▪▪▪▪▪▪▪▪▪▪▪▪▪▪▪▪

As outlined above we agree with the reviewer that we could be more clearer with our age model, and that this section could benefit from being moved into the main text. Therefore, we will make the following changes that the reviewer in these comments has suggested:

3) Age model and 14C calibration

The authors made the effort to test different approaches to establish an age model, but in the end the reader does not know, which age model/age control points were used to produce the record of the data vs. age as shown in Figure 3. So please, specify this and provide either in the main manuscript or in the supplementary material a table listing the final age control points. Did you combine? If yes, did you then discard some calibrated ages?

We agree with the reviewer that we should expand our age-depth model – in a revised MS we will add a section of the text to include a more detailed discussion and explanation of the age model.

Issues with the text and information in Table 1 regarding the 14C calibration: Table 1 and section 2.4 and supplementary material: your measured age should be the same as the conventional age, i.e. the raw 14C concentration converted into an uncorrected 14C age (using the Libby half-life). If you calibrate with Marine13 this uncorrected age would be the one used to calibrate. So I do not understand how your Table 1 can list conventional ages that are 400 years higher than the measured age –which to me looks like a reservoir age correction going into the wrong direction! And I am not sure, which age –measured or conventional– was actually calibrated! If you analyze marine material like foraminifera the measured/conventional age needs to be corrected for the reservoir effect, i.e. transferred to "atmospheric 14C levels" by subtracting the reservoir age (such as 400 yr), if you want to calibrate with atmospheric level calibration data like Intcal13. Since you are calibrating with Marine13 you do not use a fixed reservoir age (of 400 years)! During the Holocene (0-10.5 cal ka BP) section the reservoir age is provided as outcome of the ocean-atmosphere box diffusion model and varies "significantly" over time –see for example Figure 4b in Hughen et al. 2004 on Marine04. In the glacial section, where a fixed reservoir age is used, the value is 405 years and not 400 years (see p. 1877 in Reimer et al. 2013). Inconsistency between p. 3 line 30, supplementary material: you state that the deepest/oldest 14C age was not used/excluded; so why it is then shown and used in Figure S2? While correcting the 14C calibration will change the age model, this will not affect the general conclusions of the manuscript.

We thank the reviewer for noticing the discrepancy – and will alter the text accordingly.

Supplementary material text: line 24 insert $\delta$18O before ice core and mention that the NGRIP record is on the GICC05 time scale.

The data is based upon NGRIP (North Greenland Ice Core Project members et al., 2004), in a revised MS we will adjust this to GICC05.

Figure S4: the right panel does not show the filtered NGRIP record = tuning target. Why is the SPECMAP error applied and not the GICC05 errors?

We used a perceived tuning error which SPECMAP calculated – not the NGRIP error – as this would give us how the signal migrates (i.e., atmospheric signal vs. ocean signal). The error for a core will naturally be larger than the NGRIP error, as one is a slow 'responder' whereas the other is a fast 'responder'.

line 27: provide more information on the "simple filter". for which frequencies did you filter and why?

We used a filtering algorithm (~500 year time window), producing a series of filtered variants (max.; min.; mean; and so forth) of the time series to reduce the variability from a high-resolution time series (NGRIP) to one that shows the major long-term changes. This may appear counter intuitive; however, this is (i) to reduce the effect of over tuning of small-scale high frequency variation and (ii) to produce an NGRIP signal that would be similar to a down core record (i.e. a smoothed signal from a high resolution signal).

**Response to Referee 2 – Brummer et al., "*Modal shift in North Atlantic seasonality during the last deglaciation*"**

We thank the reviewer for their time and for both their general and specific comments. In the following reply, reviewer comments are in RED and our own comments are in BLACK.

My main issue with the study is that the number of analyses, i.e. specimens, per sample is too low to give a representative split up in different populations. Up to 20 specimens were picked per sample, and for quite a few samples less than that were successfully analysed. What is the risk that the split into two populations for these samples is not simply due to highly variable values that only give the impression of separate populations?

20 specimens were picked at random, for every sample. This number of specimens represents the optimum number for down core coverage with the number of specimens per isotope run on a GasBench II set-up, considering time and costs. We disagree with the reviewer that if we measured more specimens the populations would necessary coalesce into a single population, though of course because we do not know the original 'shape' of the (total) population (for instance an approximate sine such as SST when plotted as a histogram has a distribution in which there are more data at the two 'end members' than in the middle) it is difficult to assess this. Although we could perform a theoretical test to see whether this is possible. Whilst, recurrence is not proof, it is intriguing that they two populations do reoccur within the sediment, we do have single isotope data from a deeper depth down core that we can add that represents a different climatological setting. Furthermore, our inference regarding picking for pooled specimens – in which the number of specimens used as the basis of a 'mean' signal is small - would still hold (i.e., section 4.3).

Whether the populations result from 'overfitting' the mixture analysis is of course a concern. The Akaike Information Criterion (AIC; Akaike, 1974. PAST manual: https://folk.uio.no/ohammer/past/pastmanual.pdf pg. 129) is a test of best fit of the mixture model for overfitting (AIC is given in column 3 - table 2) which has a small sample correction.

Page 1 Line 24: are you suggesting the deglaciation lasted for 10 kyr?

We agree the referee that this is oddly worded, we will therefore reword for clarity. It was not our intention to imply that, instead we were referring to the approximate time ('ca.') from maximum ice sheet extent until the 'minimum' extent. We will reword as:

"This represents a shift in the timing of the main plankton bloom from late to early summer in a 'deglacial' intermediate mode that persisted from the glacial maximum until the start of the Holocene."

Line 32: many more references could be cited here to better reflective the literature. These references are all from the same lab.

We will add in more references citing other labs.

Page 2, Line 29: delete the first "and"

We will delete the repetition.

Page 3, 2.3 title: add single specimens to it to distinguish from 2.4 where the bulk analyses are described.

Will be changed.

Line 21: the pachydermas weighed >10 $\mu$g?

Whilst the specimens were not weighed –the amplitude on mass 44 correlates with the weight of a specimen, allowing us to make an informed guess of the amount of carbonate per analysis. N. pachyderma is an encrusted form and we took 250-300 µm sized specimens these allow for sufficient gas for a signal to be generated.

line 24: how many specimens/what weight were used?

Approximately 1 mg of foraminifera were used – unfortunately we did not count the number of specimens.

Line 37: "varoes"

Will change to 'varies'.

Page 4, line 20: missing year in Jonkers and Kucera

We will add the year.

Line 32: I assume these are the pooled d18O?

We will clarify: "The upper ~290 cm of core T88-3P is Holocene in age as evidenced by near uniform values of pooled specimen $\delta^{18}$O values"

Line 36: "during IRD events"

We will alter 'at IRD events' to 'during IRD events'.

Page 5, line 4: The striking bimodality is quite difficult to see, it could simply be more variation in the analyses. Why not plot the results also as histograms? And similar for the d13C results; it is not easy to see now how the variations are.

We will consider making histograms, although for the core sections 340 – 380 cm (covering the section where more than on population exists), this will result in 16 histograms for a single analysis (d18O) and for a single species, therefore it would 64 histograms in total. Unless the referee agrees that a single histogram 'lumping' the data together, in combination with figure 4, is suitable.

Additionally, why is the x-axis labelled in x time 10 4 years? This is confusing, just stick to the regular ka.

We apologise for having overlooked this error (the plotting programme added x10⁴). We will remove and change kyr to ka.

Page 6, line 7: Is 250-300 $\mu$m correct?

Yes, it appears odd to use a smaller than standard (i.e., 300-355 µm) size fraction, but *N. pachyderma* is a small species. Because it is generally a small species there is the concern that by selecting too large (therefore greater mass) or too small sized specimens we could, if size is some indicator of ecology, bias the results. Despite this reduced size it is a 'thick/heavy' species given its compact form and heavy calcification (regardless of encrustment) which produces enough weight for single $\delta^{18}$O analysis.

Line 8: were any of the sediment-trap pachydermas genetically determined?

Unfortunately, this is not possible for material sinking into deep-moored sediment traps. Generally, foraminiferal shells settling into a trap at 2.5 km water depth are free of original cellular matter for genetic analysis or found infested by bacterial and ciliates consuming any remains. Given that, the trap samples are ashed to isolate the mineral skeletons from the organic matter and leave a clean residue for isotope and chemical analysis. Whilst there has been some suggestion that variance in stable isotope value may relate to genetic factors, only recently (to our knowledge) has a protocol been develop for combined genetic and stable isotopes of small samples:

https://journals.plos.org/plosone/article?id=10.1371/journal.pone.0213282

Line 35: pachyderma is also unlikely to have lived in this meltwater; they normally stick below this relatively fresh layer.

We agree – and will refer and expand upon this in a revised MS. Here, we are referring to the spike in isotope records that occur in the literature – the so-called 'meltwater spike' in a number of papers including Berger et al. (1977; https://www.nature.com/articles/269301a0), Jones & Ruddiman (1982; https://doi.org/10.1016/0033-5894(82)90056-4 ) and so forth. We will alter the text accordingly:

"The presence of continental ice-rafted debris (IRD) down core in T88-3P, without a clear concomitant 'spike' in the $\delta^{18}$O, referred to in the literature as a 'meltwater spike' (Berger et al., 1977; Jones and Ruddiman, 1982) of either *N. pachyderma* or *G. bulloides* (Fig. 2) would suggest that the difference in $\delta^{18}$O between the two populations is dominated by temperature, consistent with previous studies showing no meltwater spike (Duplessy et al., 1996; Straub et al., 2013)."

Page 7, line 31: delete "."

We will delete this.

Page 8, line 7: the Bard, 2001 reference is missing from the References

We will add it accordingly to the reference list.

Section 4.3: the results here show that in a setting like the North Atlantic the pooled specimen analyses may be biased when not enough specimens are being used. Could you provide an estimate how many specimens would be needed to give a reliable estimate?

We thank the reviewer for bringing up this is an important point. Our results are merely showing the potential error or spread between increasing in-group numbers of pooled specimens (figure 6). The aim of pooled analysis is to average out specimen to specimen variability and produce a mean value for the core interval (time-interval) sampled that can be used as a climatological signal. Until Shackleton (1965) the amount of carbonate required for a single measurement was 4.5 mg, reducing to 1 mg and subsequently the amount required has steadily decreased as the technology has evolved. Pooled specimens therefore have steadily decreased from 100's to 10's, or less. It is our opinion that it would not be correct for us to state an exact number for a reliable estimate, as this undoubtedly will change depending on the sedimentation rate, the core, the time interval, the location, the weight limitations of the mass spectrometer (upper or lower), etc. However, if one considers the question of "when not enough specimens are being used" in fact it is not so much the total number of specimens but the proportion between the populations, if there is a single population then fewer specimens may be enough (although one would still need to account for the variance within that population).

In addition, one could argue that replicates rather than group number may be better at reducing associated biases (e.g., keeping the number in group constant and performing several replicates). What we do think, however is that pooled specimens should be considered in light of this 'hidden' variance. Therefore, in a revised MS we will expand upon this section through calculation of the how much the difference in proxy information (e.g. temperature or salinity estimates) may be.

Figure 2b: Is this 14C age of 41900 years used for the age model or not? It seems not, so then it should be deleted from the figure or indicated as such.

Whilst the date is not used for the age model because of the calibration curve's assumptions around this age, we disagree that it should be left out as it (i) has been measured and (ii) gives a general indication of the relative age of this sediment. That being said, we will alter the colour of the text to red / italic to indicate a date we did not use – but we do not find 'error' with.

Figure 5: Add headings of the different areas on top of each "column".

Thank you for this suggestion. We will add both the name, area and latitude of each trap for each 'column'.

Clim. Past Discuss.,
https://doi.org/10.5194/cp-2018-144-RC1, 2019

[Figure]

Brummer and co-authors present single specimen stable isotope measurements of polar species N. pachyderma and transitional species G. bulloides for core T88-3P in the northern mid-latitude North Atlantic. The authors deduce that two different populations of N. pachyderma existed throughout the last deglaciation and that, based on modern observations in the northern North Atlantic, these populations represent calcification during different periods of the year and thus under different environmental conditions. The study provides important new insights and merits publication in a journal like Climate of the Past. However, before the current manuscript could be accepted for publication, there are several points that need to be addressed/explained better and the inconsistencies in labeling etc. need to be corrected. So overall, I am recommending major revisions.

[Figure]

There are three major concerns that I have and which I will outline first.

1) Unimodal mode of G. bulloides and G. bulloides $\delta$13C values

The authors state that the single specimen isotope data of G. bulloides are unimodal, but give not reasoning for this statement. Subsequently, they use the unimodal distribution of G. bulloides as evidence that the two populations of N. pachyderma cannot be related to bioturbation (more on this in point 2). I would like to see some justification for declaring the G. bulloides data unimodal in the text. Whereas the $\delta$18O values show much less scatter than the N. pachyderma data, the respective $\delta$13C data show a range of 0.5‰ at some levels and I wonder, if this is not a reflection of more than one population. This statement is, however, only valid if the $\delta$13C values plotted in Figure 3 are actually correct, because G. bulloides $\delta$13C values should (mostly) be negative and the scale on the Figure is positive and has exactly the same range as for N. pachyderma.

2) Influence of bioturbation

Whereas I agree with the authors in the general sense that the occurrence of two populations cannot be explained by bioturbation, I would urge them to be more careful in those cases where one of the populations is presented by only 1 to 4 specimens. In this regard, it is essential to include an abundance record (which could be the N. pachyderma ratio record from Fig. 2) of both species in Figure 3. Since Figure 2 is presented vs. depth and Figure 3 vs. age, it is impossible for the reader to see where abundance minima of the respective species could have led to a "bias" in the single specimen isotope data (also in G. bulloides during periods of near dominance of N. pachyderma). For example, I do not perceive the argument of the unimodal mode of G. bulloides valid for the two specimens of population 2 in the third line of Table 2 [see note below on correcting column 1 of this table], if that level has already a low abundance of N. pachyderma and can thus be much more likely affected by –even if assumed minor, i.e. over 5 instead of 10 or 20 cm depth– bioturbation. In addition,

[Figure]

Figure 3 should include a plot showing the variations in the sediment rates, so that the reader can see where low sedimentation rates might have increased the chance of bioturbational mixing. Including these plots might not change the story, but provides the reader with the option to judge him/herself in which levels bioturbation might have affected the single specimen data (and to what degree) or not.

3) Age model and 14C calibration

The authors made the effort to test different approaches to establish an age model, but in the end the reader does not know, which age model/age control points were used to produce the record of the data vs. age as shown in Figure 3. So please, specify this and provide either in the main manuscript or in the supplementary material a table listing the final age control points. Did you combine? If yes, did you then discard some calibrated ages? Issues with the text and information in Table 1 regarding the 14C calibration: Table 1 and section 2.4 and supplementary material: your measured age should be the same as the conventional age, i.e. the raw 14C concentration converted into an uncorrected 14C age (using the Libby half-life). If you calibrate with Marine13 this uncorrected age would be the one used to calibrate. So I do not understand how your Table 1 can list conventional ages that are 400 years higher than the measured age –which to me looks like a reservoir age correction going into the wrong direction! And I am not sure, which age –measured or conventional– was actually calibrated! If you analyze marine material like foraminifera the measured/conventional age needs to be corrected for the reservoir effect, i.e. transferred to "atmospheric 14C levels" by subtracting the reservoir age (such as 400 yr), if you want to calibrate with atmospheric level calibration data like Intcal13. Since you are calibrating with Marine13 you do not use a fixed reservoir age (of 400 years)! During the Holocene (0-10.5 cal ka BP) section the reservoir age is provided as outcome of the ocean-atmosphere box diffusion model and varies "significantly" over time –see for example Figure 4b in Hughen et al. 2004 on Marine04. In the glacial section, where a fixed reservoir age is used, the value is 405 years and not 400 years (see p. 1877 in Reimer et al. 2013). Inconsistency

between p. 3 line 30, supplementary material: you state that the deepest/oldest 14C age was not used/excluded; so why it is then shown and used in Figure S2?

While correcting the 14C calibration will change the age model, this will not affect the general conclusions of the manuscript.

Additional comments:

Main manuscript p. 3 abundance counts: please specify a) how the % IRD was calculated; b) why a Ratio of NPS was calculated and not the more commonly used % N. pachyderma.

p. 3 Stable isotope section: please mention a) the resolution at which the single specimen measurements were done (4 cm?); b) if the N. pachyderma specimens were encrusted; c) which are the international carbonate standards used during the stable isotope analyses?

p. 3 core stratigraphy (besides comments above on 14C calibration): may be specify that you follow Reimer et al. (2013) when using $\Delta$R of 0$\pm$200 yr. line 29-30: if you keep the sentence, specify which sample was excluded (do not assume that every reader will read the supplementary material in detail). line 31-32: how many specimens of G. bulloides and G. glutinata were analyzed for the "bulk" analyses? line 35: include that the tuning was done to the $\delta$18O record of NGRIP, which, I assume, is presented on the GICC05 chronology. If you used NGRIP on GICC05, did you remember to correct the GICC05 b2k ages to BP ages (by subtracting 50 years) to make the tuned ages compatible with the calibrated 14C ages? line 36-37: you are providing information on temporal resolution and not sedimentation rates. I do not find this very informative and would like to see a figure showing the variations. Also, the sentence in its current phrasing is incomplete.

p. 4 line 4: what does IFA stand for?

p. 4 line 20: year missing for Jonkers and Kucera reference

p. 5 line 14-15: what about within glacial mixing/bioturbation?

p. 6 line 35: N. pachyderma $\delta$18O data not shown in Figure 2.

Table 1: following the recommendations of Stuiver & Reimer " Users are advised to round results to the nearest 10 yr for samples with standard deviation in the radiocarbon age greater than 50 yr".

Table 2: first column: please correct; what you are listing are not or incomplete depths. since the data itself is not shown vs. depth, it would be good to have an age column as well. Reduce the number of decimal places in the Prob and Mean columns, so that the numbers become easier to read.

Figure 3, 4, S1 etc.: in all the axis label referring to the NGRIP $\delta$18O data, replace the "SW (sea water ??)" by "ice". Provide reference for NGRIP data in figure captions.

Figure 3: as mentioned already above under point 1, correct the $\delta$13C scale for G. bulloides.

Figure S4: the right panel does not show the filtered NGRIP record = tuning target. Why is the SPECMAP error applied and not the GICC05 errors?

Supplementary material text: line 24 insert $\delta$18O before ice core and mention that the NGRIP record is on the GICC05 time scale.

line 27: provide more information on the "simple filter". for which frequencies did you filter and why?
* * *
[Figure]

Clim. Past Discuss.,
https://doi.org/10.5194/cp-2018-144-RC2, 2019

[Figure]

Review of the manuscript "Modal shift in North Atlantic seasonality during the last deglaciation" by Brummer et al. The authors present a study using single specimen isotopes on the planktonic foraminifera G. bulloides and N. pachyderma to show that during the deglaciation in the North Atlantic two different populations of pachyderma, one in spring and one in late-summer, occurred, while only one existed during the glacial and the Holocene. This variation would not have been possible to resolve using traditional pooled specimen analyses. These results suggest that these two populations are reflective of modern conditions from the present Irminger Sea further to the north, where sediment trap data for pachyderma show a similar double abundance peak. An interesting implication of the results is that when the pooled specimen record is reflecting a change of population rather than presenting the same signal, does this

imply that no deglacing warming took place in this area of the North Atlantic? This study is a very interesting application of single foraminifer analyses of stable isotopes showing the use of single foraminifer analyses, highlighting the increasing attention it receives in the literature. The manuscript is mostly clearly written and easy to follow.

My main issue with the study is that the number of analyses, i.e. specimens, per sample is too low to give a representative split up in different populations. Up to 20 specimens were picked per sample, and for quite a few samples less than that were successfully analysed. What is the risk that the split into two populations for these samples is not simply due to highly variable values that only give the impression of separate populations?

Page 1 Line 24: are you suggesting the deglaciation lasted for 10 kyr? Line 32: many more references could be cited here to better reflective the literature. These references are all from the same lab.

Page 2, Line 29: delete the first "and"

Page 3, 2.3 title: add single specimens to it to distinguish from 2.4 where the bulk analyses are described. Line 21: the pachydermas weighed >10 $\mu$g? 2.4, line 24: how many specimens/what weight were used? Line 37: "varoes"

Page 4, line 20: missing year in Jonkers and Kucera Line 32: I assume these are the pooled d18O? Line 36: "during IRD events"

Page 5, line 4: The striking bimodality is quite difficult to see, it could simply be more variation in the analyses. Why not plot the results also as histograms? And similar for the d13C results; it is not easy to see now how the variations are. Additionally, why is the x-axis labelled in x time 10 4 years? This is confusing, just stick to the regular ka.

Page 6, line 7: Is 250-300 $\mu$m correct? Line 8: were any of the sediment-trap pachydermas genetically determined? Line 35: pachyderma is also unlikely to have lived in this meltwater; they normally stick below this relatively fresh layer.

[Figure]

Page 7, line 31: delete ".".

Page 8, line 7: the Bard, 2001 reference is missing from the References Section 4.3: the results here show that in a setting like the North Atlantic the pooled specimen analyses may be biased when not enough specimens are being used. Could you provide an estimate how many specimens would be needed to give a reliable estimate?

Figure 2b: Is this 14C age of 41900 years used for the age model or not? It seems not, so then it should be deleted from the figure or indicated as such. Figure 5: Add headings of the different areas on top of each "column".

To sum up, this manuscript is very suitable for Climate of the Past using a technique that is receiving more and more application. The manuscript illustrates the opportunity of single foraminifer analyses. After the authors have especially dealt with the number of specimens used per analyses and the minor comments, I see no further issues with this manuscript being published.